# Programmable 3D snapshot microscopy with Fourier convolutional networks

## Abstract

3D snapshot microscopy enables fast volumetric imaging by capturing a 3D volume in a single 2D camera image and performing computational reconstruction. Fast volumetric imaging has a variety of biological applications such as whole brain imaging of rapid neural activity in larval zebrafish. The optimal microscope design for this optical 3D-to-2D encoding is both sample- and task-dependent, with no general solution known. Deep learning based decoders can be combined with a differentiable simulation of an optical encoder for end-to-end optimization of both the deep learning decoder and optical encoder. This technique has been used to engineer local optical encoders for other problems such as depth estimation, 3D particle localization, and lensless photography. However, 3D snapshot microscopy is known to require a highly non-local optical encoder which existing UNet-based decoders are not able to engineer. We show that a neural network architecture based on global kernel Fourier convolutional neural networks can efficiently decode information from multiple depths in a volume, globally encoded across a 3D snapshot image. We show in simulation that our proposed networks succeed in engineering and reconstructing optical encoders for 3D snapshot microscopy where the existing state-of-the-art UNet architecture fails. We also show that our networks outperform the state-of-the-art learned reconstruction algorithms for a computational photography dataset collected on a prototype lensless camera which also uses a highly non-local optical encoding.

## 1 Introduction

Volumetric imaging has been a valuable technique for measuring neural activity at single neuron resolution across the brain [2, 22, 36]. We are particularly motivated by the problem of imaging neuronal activity across the whole brain of transparent model organisms such as larval zebrafish (*Danio rerio*) at higher volume rates, to enable richer studies of neural dynamics and signal propagation. Existing approaches to whole-brain imaging involve sequential acquisition of stacks of 2D images [2], typically resulting in 0.5Hz - 2Hz volume capture rates, limited by the number of 2D depth plane acquisitions. However, since neural activity occurs at orders of magnitude faster timescales, faster volume imaging rates can provide understanding of neural computations with significantly more detail.

3D snapshot microscopy techniques can significantly speed up the imaging of a 3D sample volume using non-local optical encodings to combine information corresponding to different depth planes in the volume into a single 2D image, allowing for fast volume acquisition at the frame rate of the camera sensor followed by computational decoding to reconstruct the 3D volume [5, 3, 4, 7, 13, 16, 18, 17, 20, 27, 26, 30, 38, 37, 33]. The optimal optical design for a 3D snapshot microscope is unknown in general [5], and likely depends on the specific sample structure. Newly developed programmable microscopes with up to $10^6$ free parameters, e.g. pixels on a spatial light modulator (SLM), enable the implementation of a rich space of microscope encodings, and present the possibility of direct optimization of snapshot microscope parameters specifically for particular classes of samples and imaging tasks.

Deep learning-based decoders combined with differentiable simulations of optical encoders have enabled end-to-end optimization of both the deep learning decoder and the optical encoder. This end-to-end optical engineering approach has recently been applied to depth from defocus [6, 10, 14, 35], lensless photography [13], or particle localization microscopy [23]. However, all such applications

either use far fewer optical parameters or operate on relatively compact or local encodings due to small filters in their deep networks [6, 23, 35, 10, 14, 34, 13]. There are two barriers to optimizing more parameters and producing global versus local encodings: (1) the typical convolutional networks with small filters used by previous methods for computational reconstruction have a prior to decode locally, and therefore limit optical designs to local encodings, unlike the global PSFs used in snapshot microscopy and (2) the optimization of optical parameters involves backpropagation through both the deep learning reconstruction and a differentiable optical simulation, which is more expensive computationally for non-local optical encodings. We focus on two computational optics problems that demonstrate a superior deep network architecture that can successfully reconstruct from images encoded by non-local optical encoders and with orders of magnitude more optical parameters than previously attempted. To address the computational burden of simulating and optimizing non-local encoders, we also developed a parallel, multi-GPU optical simulation framework.

**Problem statement** We define an optical encoder as $\mathbf{M}_\phi$ parameterized by $\phi$ and a computational decoder as $\mathbf{R}_\theta$ parameterized by $\theta$. We wish to develop a reconstruction network architecture for decoding non-local encodings in 2D images $\mathbf{c}$ produced by $\mathbf{M}_\phi$ and also to enable the end-to-end optimization of $\phi$ to produce non-local encodings. We'll consider two concrete applications: (1) 3D snapshot microscopy using an SLM-based programmable microscope (Figure 1A), where we wish to optimize both the reconstruction network $\mathbf{R}_\theta$ and the optical encoding $\mathbf{M}_\phi$ based on the SLM parameters in order to image 3D samples $\mathbf{v}$ and reconstruct 3D volumes $\hat{\mathbf{v}}$, and (2) lensless photography (Figure 4), where we wish to optimize only the reconstruction network $\mathbf{R}_\theta$ to produce images of natural scenes imaged by a lensless camera $\mathbf{M}$ implementing a fixed optical encoding.

## 1.1 OUR CONTRIBUTIONS

1. We developed a large-scale, parallel, multi-GPU differentiable wave optics simulation of a programmable microscope, based on a $4f$ optical model with a phase mask ($\phi$) implemented using a spatial light modulator (SLM), described further in Appendix A.1. SLMs $\phi$ can have over $10^6$ optimizable parameters.

2. We collected a large dataset of high resolution 3D confocal volumes of zebrafish larvae for the purpose of end-to-end sample-specific optimization of the parameters of the programmable 3D snapshot microscope.

3. We introduce an efficient new reconstruction network architecture for decoding from non-local optical encoders using very large global convolutions implemented via Fourier convolutions.

4. We show that our networks outperform the state-of-the-art deep networks for both volume reconstruction and microscope parameter optimization, and for image reconstruction from lensless photography [20].

5. Our method enables, for the first time, direct end-to-end optimization of highly non-local optical encoders in the space of SLM pixels with over $10^6$ parameters. In simulation, we demonstrate the potential for significant improvements in imaging resulting from sample-specific engineering of optical encoders.

## 1.2 PRIOR WORK

Neural network architectures for computational imaging have all used convolution layers with small filters. End-to-end optimization of optical encoders have largely been performed with UNet-based architectures [6, 10, 14, 35, 13], with one method using ResNet-based architectures [23]. Such optimization has always led to local optical encodings. End-to-end optimization has never been attempted for large-field of view 3D snapshot microscopy due to the difficulty of simulating and reconstructing from non-local encoders. However, small filter convolutional deep networks have been used in a non-end-to-end manner to reconstruct volumes from 3D snapshot microscopes designed using microlens arrays [33, 34]. For photography and MRI, using deep learning combined with unrolling iterations of traditional deconvolution algorithms provides the benefits of fast amortized inference and higher quality reconstructions due to learning of structural priors [8, 9, 20, 34].

In our work, we demonstrate that convolution layers with large filters implemented efficiently in the Fourier domain enable the end-to-end learning of highly non-local optical encoders for 3D snapshot microscopy. Large convolution filters have been shown to be helpful for other computer vision

applications such as semantic segmentation and salient object detection [25, 21], and the Fourier domain parameterization of small filters has been described previously [28].

A pioneering strategy in 3D snapshot microscopy has been light field microscopy [16], which employs a microlens array at the microscope's image plane to create subimages encoding both amplitude and phase of light [16, 1]. A variety of microlens-array-based light field microscopes have been used to perform whole-brain imaging [16, 27, 37, 36, 30, 26, 7, 12, 38]. [38] optimizes the placement of microlenses, but not in an end-to-end manner. Despite variation in design, microlens-based microscopes have, to various degrees, three main limitations that can be improved: 1) blocking or scattering of light between microlenses, causing light inefficiency, 2) not making use of all pixels on the camera to encode a 3D sample, leading to inefficient compression and suboptimal reconstructions $\hat{\mathbf{v}}$, and 3) a fixed optical encoding scheme.

An alternative to using microlenses is to implement a coded detection strategy using a phase mask or diffuser to spread light broadly across the camera sensor [3, 4, 5, 13, 17, 18, 38]. The designed phase masks can be implemented either by manufacturing a custom optical element or using a programmable SLM [10, 14, 35, 13, 38, 23]. Using a programmable element allows different microscope parameters to be used for different samples.

## 2 METHODS

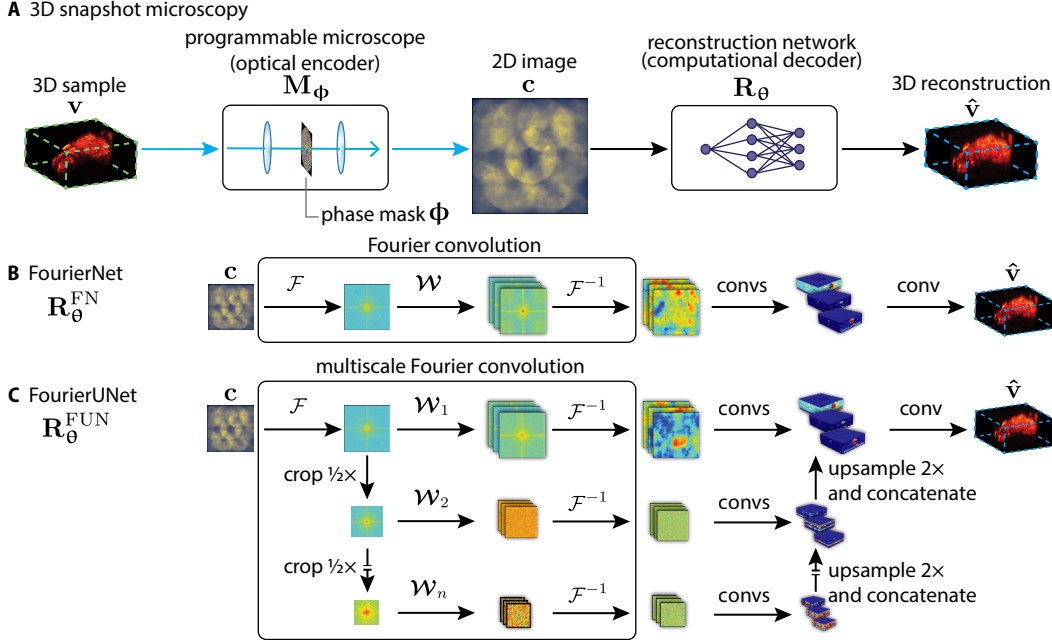

Figure 1: Overview of our problem setup and our proposed network architectures. Top row (**A**) shows the problem of 3D snapshot microscopy, where we computationally reconstruct a 3D sample from a 2D image. Middle row (**B**) shows our proposed FourierNet architecture, which includes a **Fourier convolution** layer that enables efficient computation of global features. Bottom row (**C**) shows an extension of our proposed architecture, the FourierUNet, which mimics the multiscale feature extraction of a standard UNet efficiently and with global features using a **multiscale Fourier convolution**.

We show our network architecture and an overview of autoencoder training both the microscope parameters $\phi$ and reconstruction network parameters $\theta$ in Figure 1A. The programmable microscope is simulated by a differentiable implementation of a wave-optics model of light propagation. We have selected a programmable microscope design based on pupil-plane phase modulation with a programmable spatial light modulator, for which imaging is well-approximated by a computationally-efficient convolution [11]. A detailed description of our simulation is provided in Appendix A.1. For lensless photography, there is no optical simulation because the images have been collected on a real camera.

## 2.1 FourierNet and FourierUNet architectures for reconstruction from optical encoders

Because images created by optical encoders can potentially encode signals from the incoming light field to any location in the camera image in a non-local manner, it is essential that a reconstruction network have global context. Existing multi-scale architectures such as the UNet [29] can achieve global context, but at the expense of many computation layers with small convolution kernels which we believe have a local information bias which is inappropriate for computational optics. In this paper, we introduce relatively shallow architectures for computational imaging, which rely on convolution layers with very large filters, implemented efficiently in the Fourier domain [28].

**FourierNet** We propose a simple three layer convolutional network architecture with very large global convolutions at the very first layer, followed by two standard local convolutional layers (Figure 1B). We define a global convolution as a convolution with kernel size equal to the input image. Such a convolution achieves global context in a single step but is computationally expensive. Global convolutions are implemented more efficiently in the Fourier domain, yielding a speed up of two orders of magnitude. Due to the use of Fourier convolutions to enable global context, we call our architecture FourierNet. In contrast to a typical UNet which can contain many tens of convolution layers, the FourierNet is only three layers deep, which requires backpropagation through fewer layers compared to a typical UNet with the same receptive field.

**FourierUNet** We also propose a multi-scale variant of the FourierNet by bringing together elements of the multi-scale UNet and the single-scale FourierNet. Here, we take advantage of the fact that down-sampling in image space corresponds to a simple cropping operation in the Fourier domain, resulting in a band-limited computation of a feature map. We efficiently implement multi-scale global Fourier convolutions (Figure 1C) to replace the encoding/"analysis" pathway of a UNet. We then use the standard decoding/"synthesis" pathway of the UNet to combine the multi-scale features into a single 3D volume reconstruction (Appendix A.3, A.4). Thus we can study whether multi-scale features or global context is more important for decoding non-local optical encoders.

**Fourier domain convolutions** It is well-known that large kernel convolutions can be implemented more efficiently in the Fourier domain [24, 19, 32]. A naive implementation of global convolution requires $\mathcal{O}(N^2)$ operations, where $N$ is number of pixels in both the image and the kernel. An alternative global convolution implementation is to Fourier transform the input $\mathbf{x}$ and convolution kernel $\mathbf{w}$, perform element-wise multiplication in Fourier space, and finally inverse Fourier transform, requiring only $\mathcal{O}(N \log N)$ operations [19, 32]. Following [28], we store and optimize the weights in Fourier space $\mathcal{W}$. This over-parameterization costs $8\times$ the memory of an equivalent real valued large filter but saves the computational cost of Fourier transforming the real-valued weights (Appendix A.3). Thus a Fourier convolution is defined:

$$\mathbf{Re}\{\mathcal{F}^{-1}\{\mathcal{W} \odot \mathcal{F}\{\mathbf{x}\}\}\} \tag{1}$$

For image and kernel sizes of $256 \times 256$ pixels, our implementation leads to nearly $500\times$ speedup: standard PyTorch convolution takes 2860ms, while Fourier convolution takes 5.92ms on a TITAN X.

**Multi-scale Fourier domain convolutions** It is well-known [24] that downsampling corresponds to cropping in the Fourier domain. Thus the Fourier convolution can be extended to efficiently produce multi-scale feature representations in one step (Figure 1C). We define our multi-scale Fourier convolution as

$$\left\{\mathbf{Re}\{\mathcal{F}^{-1}\{\mathcal{W_1} \odot \mathrm{crop}_1[\mathfrak{c}]\}\}, \ldots, \mathbf{Re}\{\mathcal{F}^{-1}\{\mathcal{W_n} \odot \mathrm{crop}_n[\mathfrak{c}]\}\}\right\} \tag{2}$$

where subscript denotes scale level (higher subscript indicates lower spatial scale/more cropping in Fourier space) and we precompute $\mathfrak{c} := \mathcal{F}\{\mathbf{c}\}$ once.

## 2.2 Physics-based autoencoder for simultaneous engineering of microscope encoder and reconstruction network decoder

We describe the imaging process as the following transformation from the 3D light intensity volume of the sample $\mathbf{v}$ to the 2D image formed on the camera $\mathbf{c}$:

$$\mathbf{\mu_c} = \mathbf{M_\phi}(\mathbf{v}) \tag{3}$$

$$\mathbf{c} = \max\left([\mathbf{\mu_c} + \sqrt{\mathbf{\mu_c}}\epsilon], 0\right), \epsilon \sim \mathcal{N}(0, 1) \tag{4}$$

where $\mathbf{M_\phi}$ denotes the microscope parameterized by a 2D phase mask, $\mathbf{\phi}$. This phase mask $\mathbf{\phi}$ describes the 3D-to-2D encoding of this microscope model completely. A Poisson distribution

with mean rate $\mu_{\mathbf{c}}$ describes the physics of photon detection at the camera, but sampling from this distribution is not differentiable. We approximate the noise distribution with a rectified Gaussian. We include details on $\mathbf{M}_\phi$ in Appendix A.1 [11]. Jointly training reconstruction networks and microscope parameters involves image simulation, reconstruction, then gradient backpropagation to update the reconstruction network and microscope parameters. Details on parallelization, planewise reconstruction networks, and planewise sparse gradients are provided in Appendix A.2.

Our loss function computes the normalized mean squared error (MSE) $L_{\text{HNMSE}}$ between the high pass filtered sample and reconstruction weighted by the normalized MSE $L_{\text{NMSE}}$ between the original sample and reconstruction (to include low frequency content in the objective). Formally, our loss function $L(\mathbf{v}, \hat{\mathbf{v}})$ is defined:

$$L(\mathbf{v}, \hat{\mathbf{v}}) = L_{\text{HNMSE}}(\mathbf{v}, \hat{\mathbf{v}}) + \beta L_{\text{NMSE}}(\mathbf{v}, \hat{\mathbf{v}}) \tag{5}$$

$$L_{\text{HNMSE}}(\mathbf{v}, \hat{\mathbf{v}}) = \frac{\mathbb{E}\left[(H(\mathbf{v}) - H(\hat{\mathbf{v}}))^2\right]}{\mathbb{E}(H(\mathbf{v})^2)}, L_{\text{NMSE}}(\mathbf{v}, \hat{\mathbf{v}}) = \frac{\mathbb{E}\left[(\mathbf{v} - \hat{\mathbf{v}})^2\right]}{\mathbb{E}(\mathbf{v}^2)} \tag{6}$$

where $H(\cdot)$ denotes high pass filtering and $\mathbb{E}(\cdot)$ denotes computing the mean. Both loss terms are normalized as shown to reduce variance in $L$, which can otherwise take on large magnitude values and cause training instability. For our experiments, we set the weight $\beta$ for the $L_{\text{NMSE}}$ term to 0.1.

## 3 RESULTS

**Larval Zebrafish Dataset** High resolution volumes of transgenic larval zebrafish whole brains expressing nuclear-restricted GCaMP6 calcium indicator in all neurons were imaged using a confocal microscope. These images are representative of brain-wide activity imaging. We train on 58 different zebrafish volumes (which we augment heavily) and test on 10 held-out volumes. For all experiments, we downsample the high resolution confocal data to (1.0 μm z, 1.625 μm y, 1.625 μm x). We created 4 datasets from these data called Type A, B, C, and D corresponding to imaging different fields of view. Full specifications for these datasets are in Table 4 (Appendix A.2). For Figure 2, we restrict the field of view to (200 μm z, 416 μm y, 416 μm x) with a tall cylinder cutout of diameter 193 μm and height 200 μm and image with $256 \times 256$ pixels on the simulated camera sensor. We call this setting Type D. Figure 3 and Table 2 show our larger experiments with $512 \times 512$ simulated camera pixels, with a field of view of (250 μm z, 832 μm y, 832 μm x) and sample types Type A, Type B, and Type C.

**DiffuserCam Lensless Mirflickr Dataset** We also test reconstruction performance on experimental computational photography data[1] from [20] (Figure 4). This is a dataset constructed by displaying RGB color natural images from the MIRFlickr dataset on a monitor and then capturing diffused images by the DiffuserCam lensless camera. The dataset contains 24,000 pairs of DiffuserCam and ground truth images. The goal of the dataset is to learn to reconstruct the ground truth images from the diffused images. As in [20], we train on 23,000 paired diffused and ground truth images, and test on 999 held-out pairs of images.

### 3.1 FOURIERNETS OUTPERFORM UNETS FOR ENGINEERING NON-LOCAL OPTICAL ENCODERS

Table 1: Quality of reconstructed volumes after optimizing microscope parameters to image sample Type D on $256 \times 256$ pixel camera (mean $\pm$ s.e.m., $n = 10$)

| Microscope | Reconstruction | $L_{\text{HNMSE}} \downarrow$ | MS-SSIM $\uparrow$ | PSNR $\uparrow$ | Time $\downarrow$ (s) |
|---|---|---|---|---|---|
| FourierNet2D | FourierNet3D | **0.6409 $\pm$ 0.0213** | **0.955 $\pm$ 0.004** | **34.78 $\pm$ 0.88** | **0.71** |
| FourierNet2D | FourierUNet3D | **0.6325 $\pm$ 0.0222** | **0.956 $\pm$ 0.003** | **34.74 $\pm$ 0.83** | 1.37 |
| FourierNet2D | UNet3D | 0.7659 $\pm$ 0.0130 | 0.922 $\pm$ 0.008 | 30.06 $\pm$ 0.93 | 3.68 |
| UNet2D | UNet3D | 0.7120 $\pm$ 0.0160 | 0.913 $\pm$ 0.009 | 29.17 $\pm$ 1.13 | 3.68 |

We compare optimizing microscope parameters $\phi$ with two neural networks: 1) using our FourierNet with 2D convolutions (FourierNet2D) and 2) using a vanilla UNet with 2D convolutions (UNet2D).

---

[1]Publicly available: https://waller-lab.github.io/LenslessLearning/dataset.html

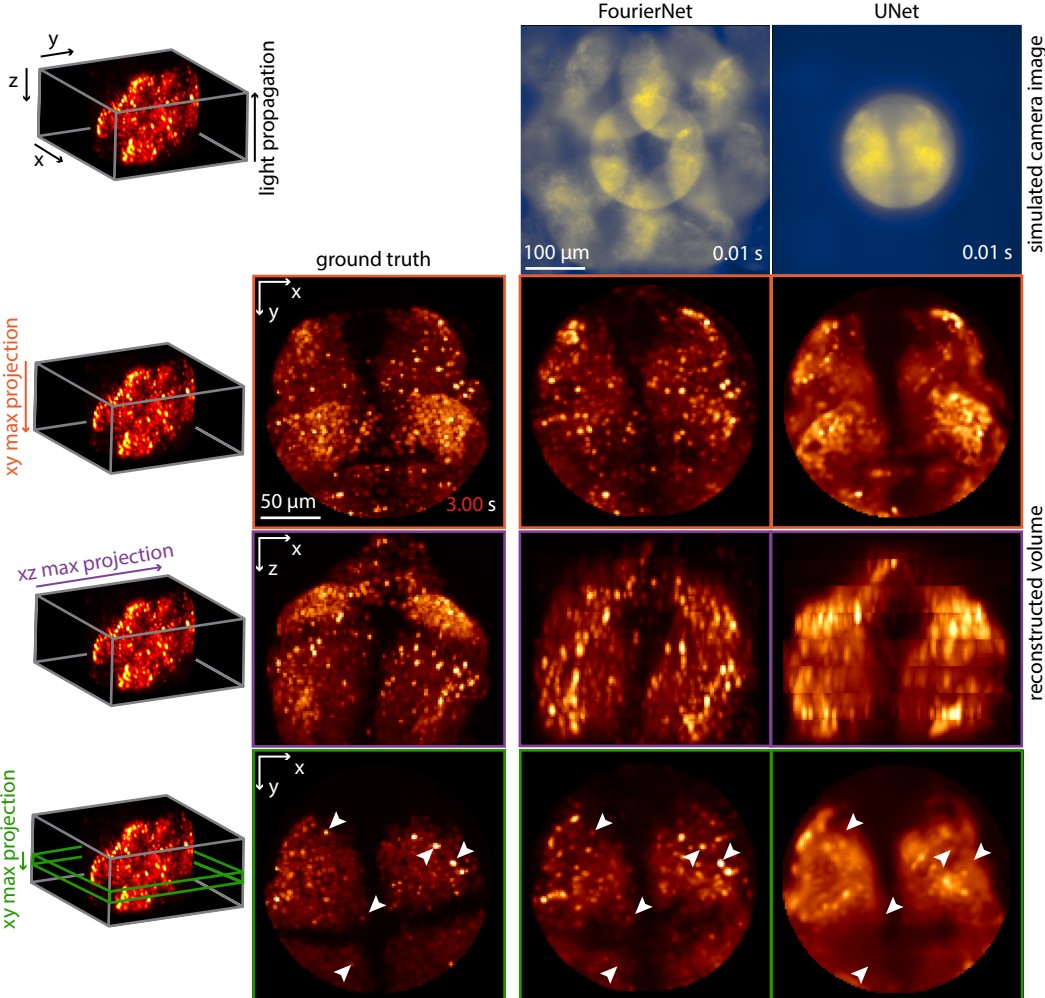

Figure 2: Comparing simulated camera images and reconstructions of a Type D volume captured using our FourierNet (middle) versus UNet (right) microscopes. Top row shows simulated $256 \times 256$ pixel camera images; bottom right of camera image shows approximate acquisition time (given a reasonable number of simulated photons per pixel, i.e. SNR).Ground truth has no corresponding camera image, because the 3D volume is imaged directly via confocal microscopy (acquisition time in xy view). Colored arrows in left column show projection axis for each row. White arrows show individual neurons clearly visible for FourierNet, but not for UNet.

Training is a two step process in which a planewise reconstruction network with fewer parameters and 2D convolutions is used during optimization of $\phi$, then once $\phi$ is fixed a more powerful reconstruction network with 3D convolutions is used. We find this scheme achieves better reconstruction quality because the reconstruction network does not need to constantly adapt to a changing optical encoding (Appendix A.3). We train these microscopes and reconstruction networks on Type D, where samples are tall cylindrical cutouts of zebrafish with diameter 193 µm and height 200 µm. Samples are imaged on a camera with $256 \times 256$ pixels (Figure 2). FourierNet2D has 2 convolution layers (with 99.8% of its kernel parameters in the initial Fourier convolution layer), while UNet2D has 32 convolution layers (kernel parameters approximately uniformly distributed per layer). FourierNet2D and UNet2D both have $\sim 4 \times 10^7$ parameters; FourierNet3D has $\sim 6 \times 10^7$ parameters vs. $\sim 10^8$ for UNet3D. Architecture details are in Appendix A.3, A.4.

Simulated camera images (Figure 2) show that the UNet microscope does not make sufficient use of camera pixels, producing only a single view of the sample. We believe this is due to a local information prior in the small kernels of UNets. The FourierNet microscope uses more camera pixels and performs better than the UNet microscope for reconstruction (Figure 2, quantified in Table 1).

Table 2: Sample specific microscope parameter optimization across 3 different zebrafish sample types imaged with $512 \times 512$ pixel camera (mean PSNR (top), MS-SSIM (bottom) $\pm$ s.e.m., $n = 10$). Green shaded regions show the regions of interest for each sample type, cropped from the full volume.

| | microscope parameters optimized for | | |
| | Type A | Type B | Type C |
| tested on |  |  |  |
|---|---|---|---|
| Type A | $\mathbf{49.75 \pm 1.35}$ | $46.01 \pm 1.33$ | $42.63 \pm 1.14$ |
| | $\mathbf{0.998 \pm 0.000}$ | $0.996 \pm 0.001$ | $0.992 \pm 0.002$ |
| Type B | $35.53 \pm 1.41$ | $\mathbf{37.28 \pm 0.96}$ | $35.34 \pm 1.16$ |
| | $0.965 \pm 0.004$ | $\mathbf{0.972 \pm 0.003}$ | $0.967 \pm 0.003$ |
| Type C | $30.87 \pm 1.15$ | $31.48 \pm 0.93$ | $\mathbf{33.79 \pm 0.90}$ |
| | $0.912 \pm 0.007$ | $0.920 \pm 0.006$ | $\mathbf{0.935 \pm 0.006}$ |

Times in Table 1 are for only the forward pass on a single GPU; one training iteration on 8 GPUs takes $\sim$0.4 seconds for FourierNet3D and $\sim$0.8 seconds for UNet3D (Appendix A.3). Both reconstruction networks must reconstruct from images that have a compressed encoding of 3D information, but the FourierNet2D is clearly more effective than the UNet2D at optimizing this encoding.

## 3.2 FOURIERNETS OUTPERFORM UNETS FOR 3D SNAPSHOT MICROSCOPY VOLUME RECONSTRUCTION

We can determine which architecture is better for volume reconstruction by choosing fixed microscope parameters and varying the architecture. In Table 1, we compare results using a FourierNet with 3D convolutions (FourierNet3D), a FourierUNet with 3D convolutions (FourierUNet3D), and a vanilla UNet with 3D convolutions (UNet3D). Architecture details are in Appendix A.3, A.4.

Reconstruction results in Table 1 compare normalized MSE $L_{\mathrm{HNMSE}}$ between the high pass filtered sample and high pass filtered reconstruction, the multiscale structural similarity $\mathrm{MS-SSIM}$ between the sample and reconstruction, and finally the peak signal-to-noise ratio PSNR. We also visualize reconstruction results for a volume in the head of a zebrafish in Figure 2. The best reconstruction networks are (equally) FourierNet3D and FourierUNet3D. The UNet3D reconstruction networks (using either microscope) fall significantly behind the FourierNet3D/FourierUNet3D reconstruction networks in all metrics.

## 3.3 ENGINEERED OPTICAL ENCODING DEPENDS ON SAMPLE STRUCTURE

To explore the effect of sample structure and size on optimized $\phi$ and the resulting reconstruction performance, we optimized $\phi$s for three different sample types: 1) Type A, samples with a short cylinder cutout of 386 µm diameter and 25 µm height, 2) Type B, samples with a tall cylinder cutout of 386 µm diameter and 250 µm height, and 3) Type C, samples without any cutout of dimension (250 µm z $\times$ 832 µm y $\times$ 832 µm x) (Table 2). All sample types were imaged with $512 \times 512$ pixels on the simulated camera. We then tested reconstruction performance on all combinations of optimized microscopes and samples, as shown in Table 2 and visualized in Figure 3 for Type B. We include architecture details in Appendix A.3, A.5.

We see in Table 2 that for all sample types, highest performance is achieved using the microscope optimized for that particular sample. These results show that our optimization process produces optical encoders that are more optimal for a specific type of sample over others.

## 3.4 FOURIERNETS OUTPERFORM STATE-OF-THE-ART FOR RECONSTRUCTING NATURAL IMAGES CAPTURED BY DIFFUSERCAM LENSLESS CAMERA

We compare our FourierNet architecture to the best learned method from [20] using an unrolled ADMM and a denoising UNet, as well as to a vanilla UNet from [20]. Architecture details are in Appendix A.3, A.6. We can see that both of our methods visually outperform the methods from [20] in Figure 4. Table 3 shows that our methods outperform the others on all metrics. In [20], a combined loss using MSE and learned perceptual loss (LPIPS) is suggested to improve visual reconstruction quality. Table 3 shows that training our FourierNet on both MSE and LPIPS results in the lowest

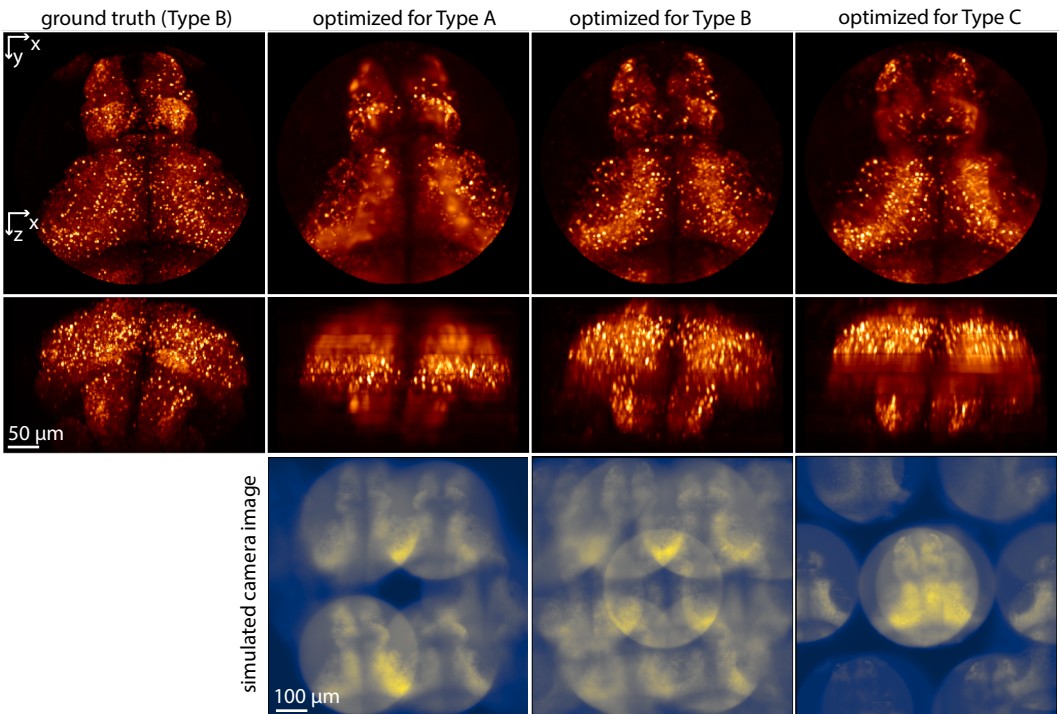

Figure 3: Reconstructed volumes resulting from imaging Type B samples by microscopes optimized for Type A, B, C. Imaging Type B samples with microscope parameters optimized for Type B samples yields the best reconstructions. Top to bottom: xy max projection, xz max projection, simulated $512 \times 512$ camera image.

Table 3: Quality of natural image reconstruction on the DiffuserCam Lensless Mirflickr Dataset (mean $\pm$ s.e.m., $n = 999$). Superscripts denote loss function: [1] MSE, [2] MSE+LPIPS.

| Method | MSE $\downarrow (\times 10^{-2})$ | LPIPS $\downarrow$ | MS-SSIM $\uparrow$ | PSNR $\uparrow$ | Time $\downarrow$ (ms) |
|---|---|---|---|---|---|
| FourierNet[1] | **0.39 $\pm$ 0.007** | 0.20 $\pm$ 0.00 | **0.882 $\pm$ 0.001** | **24.8 $\pm$ 0.09** | 37.54 |
| FourierNet[2] | 0.54 $\pm$ 0.010 | **0.16 $\pm$ 0.00** | 0.868 $\pm$ 0.001 | 23.4 $\pm$ 0.09 | 37.54 |
| FourierUNet[1] | 0.43 $\pm$ 0.009 | 0.22 $\pm$ 0.00 | 0.875 $\pm$ 0.001 | 24.5 $\pm$ 0.09 | 204.69 |
| Le-ADMM-U[2] [20] | 0.75 $\pm$ 0.021 | 0.19 $\pm$ 0.00 | 0.865 $\pm$ 0.002 | 22.1 $\pm$ 0.09 | 59.01 |
| UNet[2] [20] | 1.68 $\pm$ 0.060 | 0.24 $\pm$ 0.00 | 0.818 $\pm$ 0.002 | 19.2 $\pm$ 0.11 | **06.97** |

LPIPS loss across all methods, but does not offer better visual performance than our MSE-only model as seen in Figure 4. FourierUNet does not show improved results over FourierNet.

### 3.5 GLOBAL RECEPTIVE FIELD IS MORE IMPORTANT THAN MULTISCALE FEATURES

UNets are effective because: (1) features are computed at multiple scales and (2) large receptive fields are achieved in few layers. FourierUNets allowed us to decouple these two explanations because the receptive field is global in a single layer. We see on both our microscopy dataset (which does not have multiscale structure) and the lensless photography dataset (which does have multiscale structure) that the FourierUNet does not improve upon the FourierNet. Thus we see that it is more important for decoding from non-local optical encoders to have a global receptive field than multi-scale features.

## 4 DISCUSSION

**Summary** We have presented FourierNets for decoding from highly non-local optical encodings. We have demonstrated that FourierNets enable end-to-end optimization of non-local optical encodings

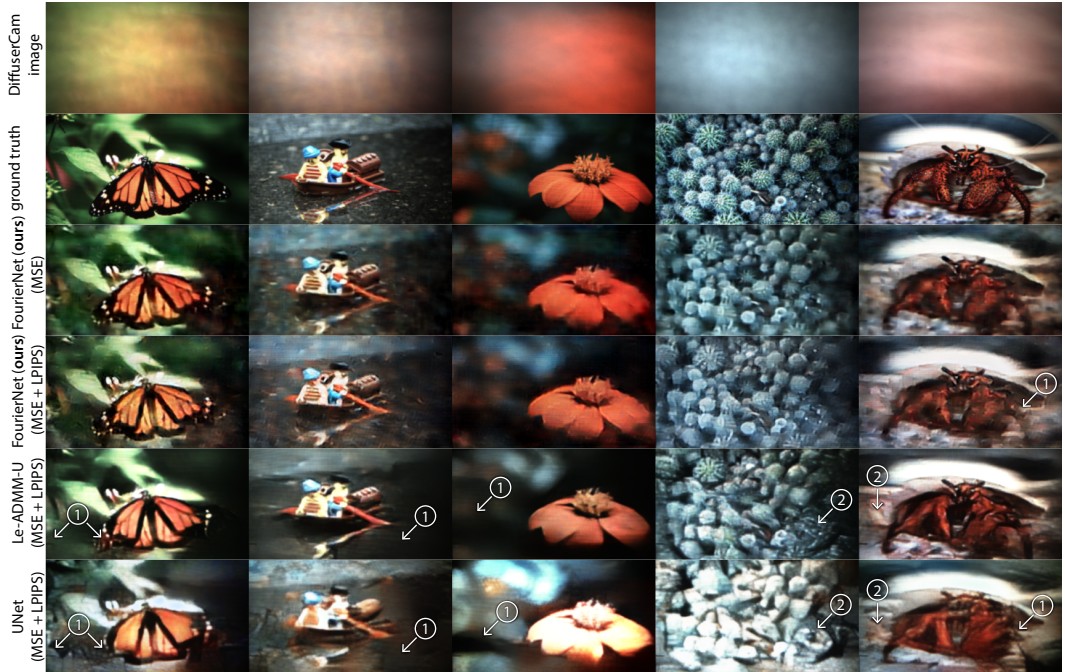

Figure 4: Comparisons of our method (second and third rows) to state-of-the-art learned reconstruction methods on lensless diffused images of natural scenes. Regions labeled ① indicate missing details, either resolution or textures in backgrounds. Regions labeled ② indicate hallucinated textures. Note that the previous state-of-the-art solutions [20] exhibit both issues more often compared to our models.

where UNets fail for simulated 3D snapshot microscopy, and also that FourierNets beat state-of-the-art decoders for lensless photography. FourierNets provide fast amortized inference and reconstruction times many orders of magnitude faster than traditional iterative reconstruction algorithms [20] and higher quality reconstructions due to effective learning of sample structural priors. Generally, our global kernel architecture using Fourier convolutions could be applied to other problems where global integration of features is necessary (though we have focused on computational microscopy and photography where such global mixing of information dominates).

**Limitations** Our engineered optical encoders have not yet been experimentally tested on a programmable microscope. This will require measuring and accounting for system aberrations and calibration to implement the engineered optical encodings. However, the parameters of even an imperfectly implemented optical encoder can be perfectly measured and used to train reconstruction networks to near optimal performance. Our claim that sample type-specific optical encoders perform better on their respective samples has only been evaluated in simulation. Further, imaging a different sample class (i.e. with different spatiotemporal statistics) will require retraining at least the reconstruction network, and ideally also the optical encoder. Our autoencoder training style requires simulation of imaging with gradients, which can be memory expensive and limits the total field of view. Our training approach requires multi-GPU parallelization (described in Appendix A.2), and in practice we optimized microscopes using 8 Quadro RTX 8000 GPUs or 4 RTX 2080 Ti GPUs for the larger and smaller experiments, respectively. Training a microscope and decoder requires at least two weeks of total training. Our Fourier convolution layers require a fixed input size to truly be global, though this could be addressed by resampling.

**Reproducibility** We train on 8 Quadro RTX 8000 GPUs for the largest experiments, and have described our pre-processing, training, and testing procedures in Appendix A.2, A.3, A.4. We will also release our simulation software, training scripts, and our datasets upon publication.

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

## A Appendix

### A.1 Forward Simulation of Programmable 3D Snapshot Microscope

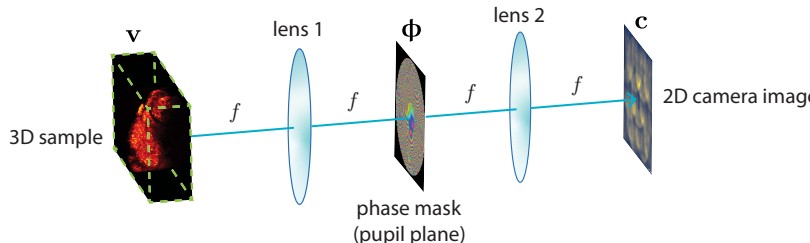

Figure 5: Diagram of a $4f$ optical model that is the basis for our simulated microscope $\mathbf{M_\phi}$, showing the Fourier plane in which we have the programmable and trainable 2D phase mask $\mathbf{\phi}$.

Here we describe our wave optics simulation of the microscope $\mathbf{M_\phi}$, which we model as a $4f$ system [11]. The $4f$ optical system consists of two lenses, the first spaced one focal length from the object plane and the second spaced one focal length away from one focal length beyond the first lens (Figure 5). In between these two lenses, we can place a phase mask to manipulate the light field before passing through the second lens and forming an image on the camera sensor.

We are concerned here with fluorescence microscopy, meaning that the sources of light that we image are individual fluorescent molecules, which we can model as point emitters. Because these molecules emit incoherent light, the camera sensor in effect sums the contributions of each point emitter. In order to model such an imaging system, we first need to address modeling a single point emitter's image on the camera.

We can analytically calculate the complex-valued light field one focal length after the first lens (which we call the pupil plane) due to a point source centered at some plane $z$ (where $z$ is a distance from the object plane $z = 0$). If the point source were centered ($x = 0, y = 0$) in the object focal plane $z = 0$, we would have a plane wave at the pupil plane, but for the more general case of a point source at an arbitrary plane $z$ relative to the object plane $z = 0$, we can analytically calculate the complex-valued light field entering the pupil plane:

$$u_{\text{point}}(\mathbf{k}; z) = \exp\left[i2\pi z \sqrt{\left(\frac{n}{\lambda}\right)^2 - ||\mathbf{k}||_2^2}\right] \qquad (7)$$

where $u_{\text{point}}$ is the incoming light field entering the pupil due to a point source centered in the plane at $z$, $\mathbf{k} \in \mathbb{R}^2$ denotes frequency space coordinates of the light field in the pupil plane, $n$ is the refractive index, and $\lambda$ is the wavelength of light [11].

In this pupil plane, we can then apply a phase mask $\mathbf{\phi}$ to the light field, which is modeled as a multiplication of $u_{\text{point}}(\mathbf{k}; z)$ and $e^{i\Phi(\mathbf{k})}$, the complex phase of the pupil function. The light field exiting the pupil is therefore described by

$$u_{\text{pupil}}(\mathbf{k}; z) = u_{\text{point}}(\mathbf{k}; z)p(\mathbf{k}) \qquad (8)$$

where $p(\mathbf{k})$ is the pupil function, composed of an amplitude $a(\mathbf{k})$ and phase $\mathbf{\phi}(\mathbf{k})$:

$$p(\mathbf{k}) = a(\mathbf{k})e^{i\Phi(\mathbf{k})} \qquad (9)$$

$$a(\mathbf{k}) = \begin{cases} 1 & ||\mathbf{k}||_2 \leq \frac{\text{NA}}{\lambda} \\ 0 & ||\mathbf{k}||_2 > \frac{\text{NA}}{\lambda} \end{cases} \qquad (10)$$

where NA is the numerical aperture of the lens [11].

The light field at the camera plane can then be described by a Fourier transform [11]:

$$\mathbf{u}_{\text{camera}} = \mathcal{F}\{\mathbf{u}_{\text{pupil}}\} \qquad (11)$$

The camera measures the intensity of this complex field:

$$s(\mathbf{x}; z) = |u_{\text{camera}}(\mathbf{x}; z)|^2 \tag{12}$$

where $\mathbf{x} \in \mathbb{R}^2$ denotes spatial coordinates in the camera plane [11].

We can call this intensity $\mathbf{s}$ the point response function (PRF). If the shape of the PRF is translationally equivariant in $\mathbf{x}$, meaning that moving a point source in-plane creates the same field at the camera, just shifted by the corresponding amount, then we call this PRF a point spread function (PSF). Note that moving the point source in $z$ will not give the same shape, which allows our system to encode depth information through the PSF [5].

In order to avoid edge effects during imaging, we simulate the PSF at a larger field of view, then crop and taper the edges of the PSF:

$$\mathbf{s}_{\text{taper}} = \text{crop}[\mathbf{s}] \odot \mathbf{t} \tag{13}$$

where $\mathbf{t}$ is a taper function created by taking the sigmoid of a distance transform divided by a width factor controlling how quickly the taper goes to 0 at the edges and $\odot$ denotes elementwise multiplication. We intentionally simulate a larger field of view than the sample in order to avoid edge artifacts. The purpose of the $\text{crop}[\cdot]$ is to cut the PSF to the correct field of view. The purpose of the tapering is to remove artifacts at the edges of the cropped PSF. After we compute this cropped and tapered PSF, we also downsample $\mathbf{s}_{\text{taper}}$ to the size of the data $\mathbf{v}$ in order to save memory.

Imaging is equivalent to the convolution of the incoming light field volume intensity $\mathbf{v}$ and the cropped and tapered PSF $\mathbf{s}_{\text{taper}}$ for a given plane. At the camera plane, the light field intensity is measured by the camera sensor. Therefore, we can describe the forward model as the following convolution and integral over planes:

$$\boldsymbol{\mu}_{\mathbf{c}}(\mathbf{x}) = \iint v(\boldsymbol{\tau}_{\mathbf{x}}; z) s_{\text{taper}}(\mathbf{x} - \boldsymbol{\tau}_{\mathbf{x}}; z) \, \mathrm{d}\boldsymbol{\tau}_{\mathbf{x}} \, \mathrm{d}z \tag{14}$$

We then model shot noise of the camera sensor to produce the final image $\mathbf{c}$, for which the appropriate model is sampling from a Poisson distribution with a mean of $\boldsymbol{\mu}_{\mathbf{c}}$ [11]:

$$\mathbf{c} \sim \text{Poisson}\left(\boldsymbol{\mu}_{\mathbf{c}}\right) \tag{15}$$

However, because we cannot use the reparameterization trick to take pathwise derivatives through the discrete Poisson distribution, we instead approximate the noise model with a rectified Gaussian distribution:

$$\epsilon \sim \mathcal{N}(0, 1) \tag{16}$$

$$\mathbf{c} \approx \max\left(\left[\boldsymbol{\mu}_{\mathbf{c}} + \sqrt{\boldsymbol{\mu}_{\mathbf{c}}}\epsilon\right], 0\right) \tag{17}$$

We now turn our attention to selecting the number of pixels used in the phase mask, i.e. the number of parameters for $\mathbf{M}_{\boldsymbol{\phi}}$. We first need to determine the pixel size for Nyquist sampling the image plane with an objective of a given NA (numerical aperture). For a given pixel size $\Delta x$, we know that in frequency space coordinates we will have a bandwidth of $\frac{1}{\Delta x}$, spanning $-\frac{1}{2\Delta x}$ to $\frac{1}{2\Delta x}$. Because we must have

$$||\mathbf{k}||_2 \leq \frac{\text{NA}}{\lambda} \tag{18}$$

we know that the Nyquist sampling pixel size is given by

$$\Delta x^* = \frac{\lambda}{2\text{NA}}. \tag{19}$$

Therefore, in the image plane, for a desired field of view $L$ we must have at least

$$N^* = \frac{L}{\Delta x^*} \tag{20}$$

pixels. The discretization in the pupil plane will be the same, which means we will need to have at least $N^*$ pixels in the pupil plane to achieve the appropriate light field in the image plane. For our settings of NA $= 0.8$, $\lambda = 0.532\mu\text{m}$, and $L = 823\mu\text{m}$, we have $\Delta x^* = 0.3325\mu\text{m}$ and $N^* = 2476$ pixels. Thus, a reasonable choice is $\Delta x = 0.325\mu\text{m}$ and $N = 2560$ pixels. Note that these simulation parameters are independent of the camera pixels; we have determined only how many pixels must be used in the phase mask in order to ensure our PSF can occupy the full field of view. The camera sensor can sample the field at the image plane at an independent pixel size.

## A.2 TRAINING PSFS AND VOLUME RECONSTRUCTION NETWORKS

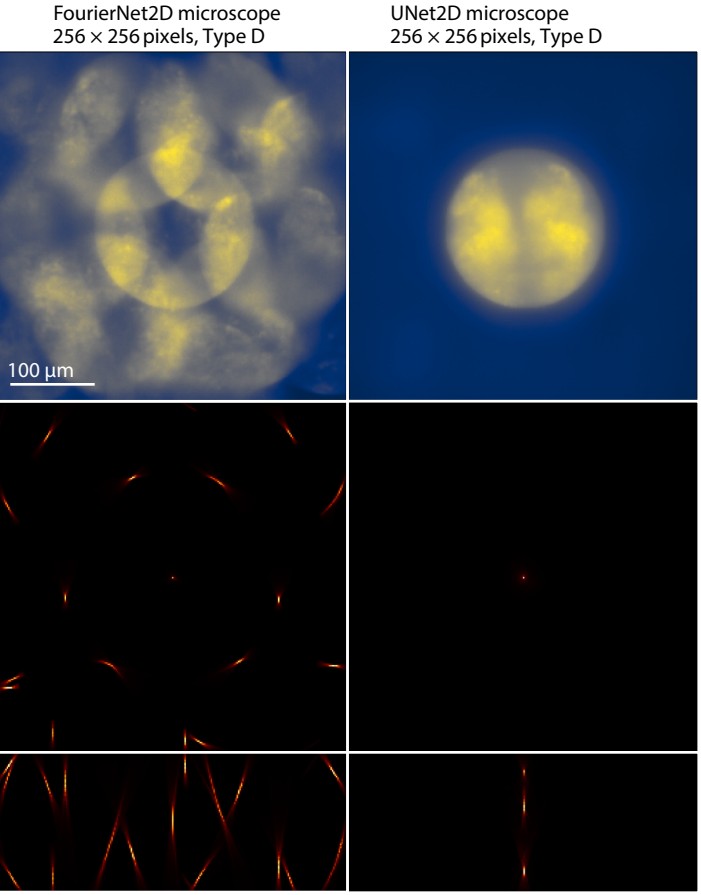

Figure 6: FourierNet successfully optimizes a PSF to image and reconstruct Type D where UNet fails. The FourierNet learned to produce multiple pencils in its PSF, which create multiple views of the sample in the camera image. UNet learned only a single pencil and fails to utilize the majority of pixels in the camera image to encode views of the sample. Top row shows simulated camera image of a Type D example, middle row shows xy max projection of the PSF, and bottom row shows xz max projection of the PSF.

Given a simulation of imaging, we can define two modes of autoencoder training: (1) jointly training the phase mask parameters $\phi$ and weak reconstruction networks in order to learn a good PSF for a particular class of samples (i.e. samples with the same spatiotemporal statistics), and (2) training a stronger reconstruction network only with a fixed, pre-trained $\phi$.

**Definition of terms** For both cases of training, the general framework is to simulate imaging using confocal volumes of pan-neuronal labeled larval zebrafish, reconstruct from the simulated image, then update the reconstruction network and, if desired, the microscope parameters. We will define the microscope parameters as $\phi$ and the reconstruction network parameters as $\theta$ for any reconstruction network $\mathbf{R}_\theta(\mathbf{c})$ where $\mathbf{R}_\theta$ maps 2D images to 3D volume reconstructions. For our training algorithms listed below, we also define: $\mathbf{D}$ our **dataset**, $\mathbf{v}$ a **ground truth volume**, $\hat{\mathbf{v}}$ a **reconstructed volume**, $L$ a computed **loss**, $z_s$ a list of $z$ plane indices that will be imaged/reconstructed, $\alpha_\phi$ the learning rate for the microscope parameters, $\alpha_\theta$ the learning rate for the reconstruction network parameters, and $\beta$ the weight of the non-high pass filtered component of the loss. When selecting a random ground truth volume, we also perform random shift, rotation, flip, and brightness augmentations.

**Microscope simulation parameters** When simulating the zebrafish imaging, we use a wavelength of 0.532 μm for all simulations. The NA of our microscope is 0.8. The refractive index $n$ is 1.33. We downsample all volumes to (1.0 μm z, 1.625 μm y, 1.625 μm x). We use a taper width of 5 for all

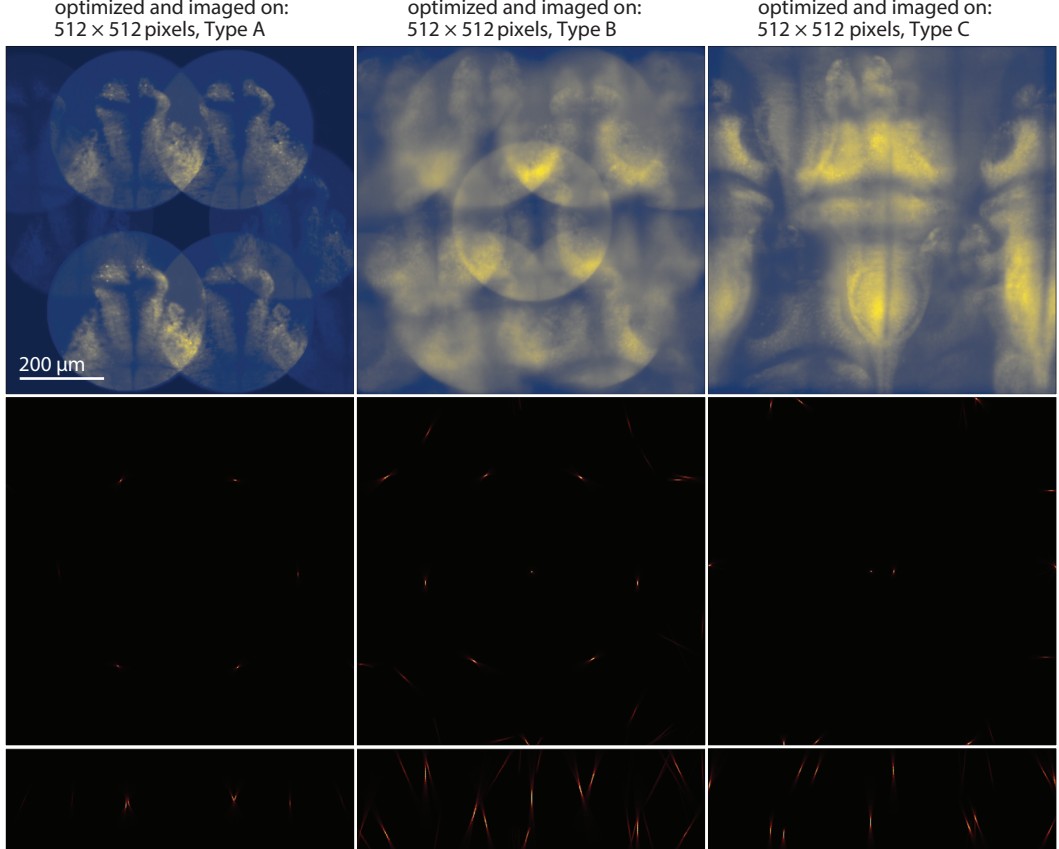

Figure 7: Optimizing PSFs for different samples result in PSFs tailored to each sample. Note that PSF optimized for Type A (left) has pencils with a span in $z$ that matches Type A. PSF optimized for Type B (middle) has pencils that span the entire $z$ depth. PSF optimized for Type C (right) has pencils spread farther apart to account for the larger sample. Top row shows simulated camera image of a Type A, B, or C example respectively, middle row shows xy max projection of the PSF, and bottom row shows xz max projection of the PSF.

simulations, and simulate the PSF at 50% larger dimensions in x and y. The resolution of the camera (for all zebrafish datasets) is also (1.625 µm y, 1.625 µm x).

**Initialization of ф** For Type A, B, and D, we initialize ф to produce a PSF consisting of 6 pencil beams at different locations throughout the depth of the volume, with the centers of these beams arranged in a hexagonal pattern in x and y. Because our optimizations generally find PSFs with many pencils, we find that initializing with such a pattern helps to converge to a more optimal PSF (data not shown).

For Type C, we instead initialize with a single helix spanning the depth of the volume (the "Potato Chip" from [5]), which seems to find a local minimum for ф that produces a PSF with more pencils (and therefore views in the camera image).

**Data settings and augmentation for Type A, B, C, D** Using our total 58 training zebrafish volumes and 10 testing zebrafish volumes (imaged through confocal microscopy), we crop in four different ways to create four different datasets. For training volumes, we crop from random locations from each volume as a form of augmentation. For testing, we crop from the same location. Physically, these crops correspond to either placing a circular aperture before light hits the $4f$ system or changing the illumination thickness in $z$, because samples would be illuminated from the side in a real implementation of this microscope. We model these by cropping cylinders (or cubes if there is no aperture) of different diameters and heights. We show details for all types Type A, B, C, D in Table 4,

where the diameter of the cylinder is labeled "Aperture Diameter" and the illumination thickness is labeled "Height".

We augment our volumes during training by taking random locations from these volumes, randomly flipping the volumes in both z and y, and also randomly rotating in pitch, yaw, and roll. Most importantly, we also randomly scale the brightness of our samples and add random background levels which serve to adjust the signal-to-noise ratio (SNR) of the resulting simulated images. The only exception to these augmentations is Type C, where we set all the volumes to the same in-plane vertical orientation (while still applying rotation augmentations in pitch and roll).

Table 4: Specifications of all zebrafish datasets Type A, B, C, D for reconstruction

| Dataset | Camera (px) | Height (planes) | Span (z, y, x) (μm) | Aperture Diameter (μm) |
|---------|-------------|-----------------|---------------------|------------------------|
| Type A | $512 \times 512$ | 12 | (25, 832, 832) | 386 |
| Type B | $512 \times 512$ | 128 | (250, 832, 832) | 386 |
| Type C | $512 \times 512$ | 128 | (250, 832, 832) | - |
| Type D | $256 \times 256$ | 96 | (200, 416, 416) | 193 |

**Parallelizing imaging and reconstruction** Furthermore, because this simulation can become too expensive in memory to fit on a single device, we generally perform the simulation, reconstruction, and loss calculation in parallel for both training modes. Therefore, any variable that has a $_s$ subscript refers to a list of chunks of that variable that will be run on each device. A $^j$ superscript indicates a particular chunk for GPU $j$. For example $z_s$ is a list of plane indices to be imaged/reconstructed, and $z_s^j$ is the $j^{\text{th}}$ chunk of plane indices that will be imaged/reconstructed on GPU $j$. We denote parallel for any operations that are performed in parallel and scatter for splitting data into chunks and spreading across multiple GPUs. Imaging can be cleanly parallelized: chunks of a PSF and sample can be partially imaged on multiple GPUs independently because the convolution occurs per plane, then finally all partial images can be summed together onto a single GPU. The reconstructions can similarly take the final image and reconstruct partial chunks (as well as calculate losses on partial chunks) of the volume independently per device. We implicitly gather data to the same GPU when computing sums ($\sum$) or means ($\mathbb{E}$). The functions parallel image and compute PSF follow the definitions above in equations 14 and 12. In the algorithms shown here, parallel image applies the same convolution described above in equation 14.

**Sparse gradients and downsampling** We additionally support training and reconstructing only some of the planes for imaging and potentially a different partial group of planes during reconstruction, as a way to sparsely compute gradients for optimization of $\theta_m$ and save memory. The planes not imaged with gradients can still contribute to the image (without their gradients being tracked) in order to make the problem more difficult for the reconstruction network. Over multiple iterations, this can become equivalent to the more expensive densely computed gradient method, essentially trading training time for memory. An additional memory saving measure not written in the algorithms is to compute the PSF at a high resolution, then downsample the PSF using a 2D sum pool to preserve total energy in order to reduce memory usage when performing the imaging and reconstruction. We denote **with no gradient tracking** to show an operation without gradients.

## A.3 IMPLEMENTATION DETAILS

**Fourier convolution details** Our Fourier convolution uses complex number weights, implemented as two channels of real numbers. Furthermore, in order to prevent the convolution from wrapping around the edges, we have to pad the input to double the size. The size of the weight must match the size of this padded input. This means that the number of parameters for our Fourier convolution implementation is $8\times$ the number of parameters required for a global kernel in a spatial convolution (though the Fourier convolution is significantly faster). We do this to save an extra padding and Fourier operation, trading memory for speed. Because the simulation of imaging requires more memory than the reconstruction network, we found this to be an acceptable tradeoff.

**Common network details** All convolutions (including Fourier convolutions) use "same" padding. For FourierUNets and vanilla UNets, downsampling and upsampling is performed only in the $x$ and $y$

---

**Algorithm 1:** Parallel PSF engineering by joint training of reconstruction network and phase mask. Microscope $\mathbf{M_\phi}$ parameters are $\mathbf{\phi}$, reconstruction network $\mathbf{R_\theta}$ parameters are $\mathbf{\theta}$, dataset is $\mathbf{D}$, learning rates for $\mathbf{\phi}$ and $\mathbf{\theta}$ are $\alpha_\mathbf{\phi}$ and $\alpha_\mathbf{\theta}$ respectively, plane indices to image and reconstruct from $z_s$, and weight for $L_{\mathrm{NMSE}}$ is $\beta$.

---

```
Input : M_φ, φ, α_φ, R_θ, θ, α_θ, D, z_s, β
```

1 **for** $\mathbf{v} \in \mathbf{D}$ **do**
    *// select plane indices to be imaged with and without gradients*
2     $z_{s,\mathrm{gradient}}, z_{s,\mathrm{no\ gradient}} \leftarrow$ **select planes**$(z_s)$
    *// move sample planes to be imaged with gradients to multiple GPUs*
3     $\mathbf{v}_{s,\mathrm{gradient}} \leftarrow$ **scatter**$(\mathbf{v}, z_{s,\mathrm{gradient}})$
    *// move sample planes to be imaged without gradients to multiple GPUs*
4     $\mathbf{v}_{s,\mathrm{no\ gradient}} \leftarrow$ **scatter**$(\mathbf{v}, z_{s,\mathrm{no\ gradient}})$
    *// compute PSF with gradients on multiple GPUs*
5     $\mathbf{s}_{s,\mathrm{gradient}} \leftarrow$ **parallel(compute PSF**$(\mathbf{M_\phi}, z_s^j)$ **for** $z_s^j$ **in** $z_{s,\mathrm{gradient}}$**)**
    *// compute partial image with gradients on multiple GPUs*
6     $\mathbf{c}_{\mathrm{gradient}} \leftarrow$ **parallel image**$(\mathbf{s}_{s,\mathrm{gradient}}, \mathbf{v}_{s,\mathrm{gradient}})$
    *// compute PSF without gradients on multiple GPUs*
7     **with no gradient tracking**
8        $\mathbf{s}_{s,\mathrm{no\ gradient}} \leftarrow$ **parallel(compute PSF**$(\mathbf{M_\phi}, z_s^j)$ **for** $z_s^j$ **in** $z_{s,\mathrm{no\ gradient}}$**)**
9     **end**
    *// compute partial image without gradients on multiple GPUs*
10     **with no gradient tracking**
11        $\mathbf{c}_{\mathrm{no\ gradient}} \leftarrow$ **parallel image**$(\mathbf{s}_{s,\mathrm{no\ gradient}}, \mathbf{v}_{s,\mathrm{no\ gradient}})$
12     **end**
    *// compute full image by summing partial images onto one GPU*
13     $\mathbf{c} \leftarrow \sum[\mathbf{c}_{\mathrm{gradient}}, \mathbf{c}_{\mathrm{no\ gradient}}]$
    *// select plane indices to be reconstructed*
14     $z_{s,\mathrm{reconstruct}} \leftarrow$ **select planes**$(z_s)$
    *// move sample planes that will be reconstructed to multiple GPUs*
15     $\mathbf{v}_{s,\mathrm{reconstruct}} \leftarrow$ **scatter**$(\mathbf{v}, z_{s,\mathrm{reconstruct}})$
    *// compute mean of high passed sample for loss normalization*
16     $\mu_{H(\mathbf{v})} \leftarrow \mathbb{E}(H(\mathbf{v}_{s,\mathrm{reconstruct}})^2)$
    *// compute mean of sample for loss normalization*
17     $\mu_{\mathbf{v}} \leftarrow \mathbb{E}(\mathbf{v}_{s,\mathrm{reconstruct}}^2)$
    *// move reconstruction networks to multiple GPUs*
18     $\mathbf{R}_{\mathbf{\theta},s} \leftarrow$ **scatter**$(\mathbf{R_\theta})$
    *// compute reconstruction and loss on multiple GPUs*
19     $L \leftarrow$ **parallel reconstruct/loss**$(\mathbf{c}, \mathbf{v}_{s,\mathrm{reconstruct}}, \mathbf{R}_{\mathbf{\theta},s}, \mu_{H(\mathbf{v})}, \mu_{\mathbf{v}}, \beta)$
    *// compute gradients for all parameters*
20     $g_\mathbf{\theta} \leftarrow \nabla_\mathbf{\theta} L$
21     $g_\mathbf{\phi} \leftarrow \nabla_\mathbf{\phi} L$
    *// update all parameters*
22     $\mathbf{\theta} \leftarrow$ **ADAM**$(\alpha_\mathbf{\theta}, \mathbf{\theta}, g_\mathbf{\theta})$
23     $\mathbf{\phi} \leftarrow$ **ADAM**$(\alpha_\mathbf{\phi}, \mathbf{\phi}, g_\mathbf{\phi})$
24 **end**

---

dimensions (we do not downsample or upsample in $z$ because there could potentially not be enough planes to do so). We train all networks using the ADAM optimizer with all default PyTorch parameters except the learning rate, which we always set to $10^{-4}$ for the reconstruction network parameters $\mathbf{\theta}$ and $10^{-2}$ for the phase mask parameters $\mathbf{\phi}$.

**Algorithm 2:** Parallel training a reconstruction network given a pre-trained phase mask. Microscope $\mathbf{M_\phi}$ parameters are $\phi$ (phase mask), reconstruction network $\mathbf{R_\theta}$ parameters are $\theta$, dataset is $\mathbf{D}$, learning rates for $\phi$ and $\theta$ are $\alpha_\phi$ and $\alpha_\theta$ respectively, plane indices to image and reconstruct from are $z_s$, and weight for $L_{\text{NMSE}}$ is $\beta$.

---

**Input :**$\mathbf{M_\phi}, \phi, \alpha_\phi, \mathbf{R_\theta}, \theta, \alpha_\theta, \mathbf{D}, z_s, \beta$

    // compute PSF without gradients on multiple GPUs

1 **with no gradient tracking**

2   |  $\mathbf{s}_{\text{no gradient}} \leftarrow$**parallel(compute PSF($\mathbf{M_\phi}, z_s^j$) for $z_s^j$ in $z_s$)**

3 **end**

4 **for** $\mathbf{v} \in \mathbf{D}$ **do**

       // select plane indices to be imaged without gradients

5     $z_{s,\text{no gradient}} \leftarrow$ **select planes($z_s$)**

       // move sample planes to be imaged without gradients to
         multiple GPUs

6     $\mathbf{v}_{s,\text{no gradient}} \leftarrow$ **scatter($\mathbf{v}, z_{s,\text{no gradient}}$)**

       // move necessary PSF planes to multiple GPUs

7     $\mathbf{s}_{s,\text{no gradient}} \leftarrow$ **scatter($\mathbf{s}_{\text{no gradient}}, z_{s,\text{no gradient}}$)**

       // compute image without gradients on multiple GPUs

8     **with no gradient tracking**

9     |  $\mathbf{c} \leftarrow$ **parallel image($\mathbf{s}_{s,\text{no gradient}}, \mathbf{v}_{s,\text{no gradient}}$)**

10     **end**

       // select plane indices to be reconstructed

11     $z_{s,\text{reconstruct}} \leftarrow$ **select planes($z_s$)**

       // move sample planes that will be reconstructed to multiple
         GPUs

12     $\mathbf{v}_{s,\text{reconstruct}} \leftarrow$ **scatter($\mathbf{v}, z_{s,\text{reconstruct}}$)**

       // compute mean of high passed sample for loss normalization

13     $\mu_{H(\mathbf{v})} \leftarrow \mathbb{E}[H(\mathbf{v}_{s,\text{reconstruct}})^2]$

       // compute mean of sample for loss normalization

14     $\mu_{\mathbf{v}} \leftarrow \mathbb{E}[\mathbf{v}_{s,\text{reconstruct}}^2]$

       // move reconstruction networks to multiple GPUs

15     $\mathbf{R}_{\theta,s} \leftarrow$ **scatter($\mathbf{R_\theta}$)**

       // compute reconstruction and loss on multiple GPUs

16     $L \leftarrow$ **parallel reconstruct/loss($\mathbf{c}, \mathbf{v}_{s,\text{reconstruct}}, \mathbf{R}_{\theta,s}, \mu_{H(\mathbf{v})}, \mu_{\mathbf{v}}, \beta$)**

       // compute gradients for reconstruction networks only

17     $g_\theta \leftarrow \nabla_\theta L$

       // update reconstruction network parameters only

18     $\theta \leftarrow$**ADAM($\alpha_\theta, \theta, g_\theta$)**

19 **end**

---

**Algorithm 3:** Parallel imaging. PSF planes on multiple GPUs are $\mathbf{s}_s$, sample planes on multiple GPUs to be imaged are $\mathbf{v}_s$.

---

**Input**   :$\mathbf{s}_s, \mathbf{v}_s$

**Output :**$\mathbf{c}$

// compute images in parallel on multiple GPUs, then sum to
   single GPU

1 $\mathbf{c} \leftarrow \sum[\textbf{parallel(convolve($\mathbf{s}_\mathbf{s}^\mathbf{j}, \mathbf{v}_\mathbf{s}^\mathbf{j}$) for ($\mathbf{s}_s^j, \mathbf{v}_\mathbf{s}^\mathbf{j}$) in ($\mathbf{s}_s, \mathbf{v}_s$))}]$

2 **return** $\mathbf{c}$

---

**Normalization** We use **input scaling** during both training and inference in order to normalize out differences in the brightness of the image and prevent instabilities in our gradients. This means we divide out the median value of the input (scaled by some factor in order to bring the loss to a reasonable range) and then undo this scaling after the output of the network. This effectively linearizes our reconstruction networks, meaning a scaling of the image sent to the network will exactly scale the output by that value. We also find this is a more effective and simpler alternative to

**Algorithm 4:** Parallel reconstruction/loss calculation. Camera image is $\mathbf{c}$, sample planes on multiple GPUs are $\mathbf{v}_s$, reconstruction networks on multiple GPUs are $\mathbf{R}_{\theta,s}$, mean for $L_{\text{HNMSE}}$ normalization is $\mu_{H(\mathbf{v})}$, mean for $L_{\text{NMSE}}$ normalization is $\mu_{\mathbf{v}}$, and weight for $L_{\text{NMSE}}$ is $\beta$.

---

**Input** : $\mathbf{c}, \mathbf{v}_s, \mathbf{R}_{\theta,s}, \mu_{H(\mathbf{v})}, \mu_{\mathbf{v}}, \beta$
**Output** : L
// compute reconstruction and loss in parallel on multiple GPUs
1 $\hat{\mathbf{v}}_s \leftarrow$ `concatenate(parallel(`$\mathbf{R}_s^j(\mathbf{c})$ `for` $\mathbf{R}_s^j$ `in` $\mathbf{R}_{\theta,s}$`))`
2 $L_s \leftarrow$ `parallel(`$\frac{\mathbb{E}[(H(\mathbf{v}_s^j) - H(\hat{\mathbf{v}}_s^j))^2]}{\mu_{H(\mathbf{v})}} + \beta \frac{\mathbb{E}[(\mathbf{v}_s^j - \hat{\mathbf{v}}_s^j)^2]}{\mu_{\mathbf{v}}}$ `for` $(\hat{\mathbf{v}}_s^j, \mathbf{v}_s^j)$ `in` $(\hat{\mathbf{v}}_s, \mathbf{v}_s)$`)`
// compute mean of scattered losses on single GPU
3 $L \leftarrow \mathbb{E}[L_s]$
4 `return` $L$

---

using a BatchNorm on our inputs. We continue to use BatchNorm between our convolution layers within the reconstruction network [15], which is effectively InstanceNorm in our case where batch size is 1 [31].

**Planewise network training logic** When we train PSFs by optimizing $\phi$, we train separate reconstruction networks per plane. This allows us to flexibly compute sparse gradients across different planes from iteration to iteration, as described in Appendix A.2. In order to do this, we create placeholder networks on any number of GPUs, then copy the parameters stored on CPU for each plane's reconstruction network to a network on the GPU as needed during a forward pass. After calculating an update with the optimizer, we copy the parameter values back to the corresponding parameter on CPU.

**Training times** We optimize our Type D microscopy experiments on 8 RTX 2080 Ti GPUs. For these, we can compare training times for the different network architectures. One training iteration (including microscope simulation, reconstruction, backpropagation, and parameter update) takes $\sim$0.6 seconds for FourierNet2D and $\sim$1.3 seconds for UNet2D when optimizing both $\phi$ and $\theta$. One training iteration takes $\sim$0.4 seconds for FourierNet3D, $\sim$0.7 seconds for FourierUNet3D, and $\sim$0.8 seconds for UNet3D when only optimizing $\theta$. More details are found in Tables 5 and 6.

Table 5: Type D experiment training times

| Network | Optimizing | # parameters | # train steps | Train step time (s) | Total time (h) |
|---|---|---|---|---|---|
| FourierNet2D | $\theta, \phi$ | $\sim 4.2 \times 10^7$ | $10^6$ | $\sim 0.8$ | $\sim 222$ |
| FourierNet3D | $\theta$ | $\sim 6.3 \times 10^7$ | $10^6$ | $\sim 0.4$ | $\sim 111$ |
| FourierUNet3D | $\theta$ | $\sim 8.4 \times 10^7$ | $10^6$ | $\sim 0.7$ | $\sim 194$ |
| UNet2D | $\theta, \phi$ | $\sim 4.0 \times 10^7$ | $10^6$ | $\sim 1.3$ | $\sim 361$ |
| UNet3D | $\theta$ | $\sim 1.0 \times 10^8$ | $10^6$ | $\sim 0.8$ | $\sim 222$ |

A.4 DETAILS FOR FOURIERNETS OUTPERFORM UNETS FOR ENGINEERING NON-LOCAL OPTICAL ENCODERS AND 3D SNAPSHOT MICROSCOPY VOLUME RECONSTRUCTION

For our experiments in Sections 3.1 and 3.2, we use 40 planes at $5\mu$m resolution in z and therefore 40 reconstruction networks to train PSFs. When training reconstruction networks only to produce the higher quality reconstructions, we use 96 planes at $1\mu$m resolution in z (chosen so that the planes actually span 200 $\mu$m in z). We train in both settings without any sparse planewise gradients, meaning we image and reconstruct all 40 or all 96 planes, respectively. We show details of all datasets used for training reconstructions in Table 4.

We show the details of our FourierNet2D architecture for training PSFs in Table 7 and our FourierNet3D architecture for training reconstruction networks in Table 8. We also show details for training times for both training PSFs and for training more powerful reconstruction networks in Table 5. We

Table 6: Type A, B, C experiment training times

| Network | Optimizing | # parameters | Type | # train steps | Train step time (s) | Total time (h) |
|---|---|---|---|---|---|---|
| FourierNet2D | $\theta, \phi$ | $\sim 1.7 \times 10^8$ | A | $5.8 \times 10^5$ | $\sim 1.1$ | $\sim 177$ |
| FourierNet3D | $\theta$ (fixed $\phi$ for A) | $\sim 3.4 \times 10^8$ | A | $\sim 2.6 \times 10^5$ | $\sim 1.6$ | $\sim 116$ |
| FourierNet3D | $\theta$ (fixed $\phi$ for A) | $\sim 3.4 \times 10^8$ | B | $\sim 1.3 \times 10^5$ | $\sim 1.6$ | $\sim 58$ |
| FourierNet3D | $\theta$ (fixed $\phi$ for A) | $\sim 3.4 \times 10^8$ | C | $\sim 1.3 \times 10^5$ | $\sim 1.6$ | $\sim 58$ |
| FourierNet2D | $\theta, \phi$ | $\sim 1.7 \times 10^8$ | B | $5.8 \times 10^5$ | $\sim 1.1$ | $\sim 177$ |
| FourierNet3D | $\theta$ (fixed $\phi$ for B) | $\sim 3.4 \times 10^8$ | A | $\sim 1.2 \times 10^5$ | $\sim 1.6$ | $\sim 53$ |
| FourierNet3D | $\theta$ (fixed $\phi$ for B) | $\sim 3.4 \times 10^8$ | B | $10^6$ | $\sim 1.6$ | $\sim 444$ |
| FourierNet3D | $\theta$ (fixed $\phi$ for B) | $\sim 3.4 \times 10^8$ | C | $\sim 5.0 \times 10^5$ | $\sim 1.6$ | $\sim 222$ |
| FourierNet2D | $\theta, \phi$ | $\sim 1.7 \times 10^8$ | C | $5.8 \times 10^5$ | $\sim 1.1$ | $\sim 177$ |
| FourierNet3D | $\theta$ (fixed $\phi$ for C) | $\sim 3.4 \times 10^8$ | A | $\sim 3.4 \times 10^5$ | $\sim 1.6$ | $\sim 151$ |
| FourierNet3D | $\theta$ (fixed $\phi$ for C) | $\sim 3.4 \times 10^8$ | B | $\sim 3.4 \times 10^5$ | $\sim 1.6$ | $\sim 151$ |
| FourierNet3D | $\theta$ (fixed $\phi$ for C) | $\sim 3.4 \times 10^8$ | C | $\sim 3.7 \times 10^5$ | $\sim 1.6$ | $\sim 164$ |

Table 7: FourierNet2D detailed architecture (1 per plane)

| Layer type | Kernel size | Stride | Notes | Shape (C, D, H, W) |
|---|---|---|---|---|
| InputScaling | - | - | scale: 0.01 | (1, 1, 256, 256) |
| FourierConv2D | (256, 256) | (2, 2) | - | (8, 1, 256, 256) |
| LeakyReLU | - | - | slope: -0.01 | (8, 1, 256, 256) |
| BatchNorm2D | - | - | - | (8, 1, 256, 256) |
| Conv2D | (11, 11) | (1, 1) | - | (1, 1, 256, 256) |
| ReLU | - | - | - | (1, 1, 256, 256) |
| InputRescaling | - | - | scale: 0.01 | (1, 1, 256, 256) |

Table 8: FourierNet3D detailed architecture (8 GPUs)

| Layer type | Kernel size | Stride | Notes | Shape (C, D, H, W) |
|---|---|---|---|---|
| InputScaling | - | - | scale: 0.01 | (1, 1, 256, 256) |
| FourierConv2D | (256, 256) | (2, 2) | - | (60, 1, 256, 256) |
| LeakyReLU | - | - | slope: -0.01 | (60, 1, 256, 256) |
| BatchNorm2D | - | - | - | (60, 1, 256, 256) |
| Reshape2D3D | - | - | - | (5, 12, 256, 256) |
| Conv3D | (11, 7, 7) | (1, 1, 1) | - | (5, 12, 256, 256) |
| LeakyReLU | - | - | slope: -0.01 | (5, 12, 256, 256) |
| BatchNorm3D | - | - | - | (5, 12, 256, 256) |
| Conv3D | (11, 7, 7) | (1, 1, 1) | - | (1, 12, 256, 256) |
| ReLU | - | - | - | (1, 12, 256, 256) |
| InputRescaling | - | - | scale: 0.01 | (1, 12, 256, 256) |

trained all networks for Type D for the same number of iterations (more than necessary for PSFs to meaningfully converge)[2].

The architecture of FourierUNet3D is 4 scales, with a cropping factor of 2 per scale in the encoding path and an upsampling factor of 2 in the decoding path. For each scale, we perform a Fourier convolution in the encoding path producing 480 feature maps, which are concatenated with the incoming feature maps of the decoding convolutions at the corresponding scale (just as in a normal UNet). In the decoding path, we use 3D convolutions with kernel size (3, 5, 5), producing 12 3D feature maps each. There are two such convolutions per scale. Note that this requires we reshape the 2D feature maps from the Fourier convolutions to 3D. This is followed by a 1x1 convolution

---

[2]Training times are approximate, and actual total time was longer due to checkpointing/snapshotting/validation of data and/or differences in load on the clusters being used.

Table 9: FourierUNet3D detailed architecture (8 GPUs)

| Scale | Repeat | Layer type | Kernel size | Stride | Notes | Shape (C, D, H, W) |
|---|---|---|---|---|---|---|
| 1 | 1 | InputScaling | - | - | scale: 0.01 | (1, 1, 256, 256) |
| 1 | 1 | Multiscale FourierConv2D + ReLU + BatchNorm2D | (256, 256) | (2, 2) | - | (60, 1, 256, 256) |
| 2 | | | (128, 128) | (2, 2) | | (60, 1, 128, 128) |
| 3 | | | (64, 64) | (2, 2) | | (60, 1, 64, 64) |
| 4 | | | (32, 32) | (2, 2) | | (60, 1, 32, 32) |
| 4 | 1 | Reshape2D3D | - | - | - | (5, 12, 32, 32) |
| 3 | 1 | Upsample2D | - | - | - | (5, 12, 64, 64) |
| 3 | 2 | Conv3D + ReLU + BatchNorm3D | (11, 7, 7) | (1, 1, 1) | - | (5, 12, 64, 64) |
| 2 | 1 | Upsample2D | - | - | - | (5, 12, 128, 128) |
| 2 | 2 | Conv3D + ReLU + BatchNorm3D | (11, 7, 7) | (1, 1, 1) | - | (5, 12, 128, 128) |
| 1 | 1 | Upsample2D | - | - | - | (5, 12, 256, 256) |
| 1 | 2 | Conv3D + ReLU + BatchNorm3D | (11, 7, 7) | (1, 1, 1) | - | (5, 12, 256, 256) |
| 1 | 1 | Conv3D + ReLU | (1, 1, 1) | (1, 1, 1) | - | (1, 12, 256, 256) |
| 1 | 1 | InputRescaling | - | - | scale: 0.01 | (1, 12, 256, 256) |

producing the 3D reconstruction output. We show a diagram of this architecture in Figure 1C, and details of this architecture in Table 9.

For our UNet2D, each encoding convolution produced 24 feature maps (except the first scale, for which the first convolution produced 12 feature maps and the second convolution produced 24 feature maps). Each decoding convolution produced 24 feature maps, but took an input of 48 feature maps where 24 feature maps were concatenated from the corresponding encoding convolution at that scale. At the end of the UNet2D, a (1, 1) convolution reduced the 24 final feature maps to 1 feature map. This single feature map is interpreted as the final output of the network, i.e. the reconstructed plane. UNet2D requires many more feature maps per plane and more layers than FourierNet, because these are necessary in order for the network to be able to integrate information from a larger field of view. We show the details of our UNet2D architecture in Table 10.

The architecture of the vanilla UNet3D is also 4 scales, with a max pooling factor of 2 per scale in the encoding path and an upsampling factor of 2 in the decoding path. Each scale of the encoding path produces 480 2D feature maps. These are concatenated to the incoming feature maps of the decoding convolutions at the corresponding scale, again with a reshape from 2D to 3D. Each scale of the decoding path produces 48 3D feature maps. Again, this is followed by a 1x1 convolution producing the 3D reconstruction output. All convolutions are in 3D with a kernel size of (5, 7, 7), with the $z$ dimension being ignored for the encoding path because the input is 2D. We show the details of our UNet3D architecture in Table 11.

### A.5 DETAILS FOR ENGINEERED OPTICAL ENCODING DEPENDS ON SAMPLE STRUCTURE

For our experiments in Section 3.3, we use 64 planes at $1\mu m$ resolution in z and therefore 64 reconstruction networks to train PSFs. When training reconstruction networks only to produce the higher quality reconstructions, we use 128 planes at $1\mu m$ resolution in z (chosen so that the planes actually span 250 $\mu m$ in z). We train in the reconstruction only setting without any sparse planewise gradients, meaning we image and reconstruct all 128 planes. However, when training a PSF we image and reconstruct 40 planes at a time with gradient per iteration (spread across 8 GPUs). These 40

Table 10: UNet2D detailed architecture (1 per plane)

| Scale | Repeat | Layer type | Kernel size | Stride | Notes | Shape (C, D, H, W) |
|---|---|---|---|---|---|---|
| 1 | 1 | InputScaling | - | - | scale: 0.01 | (1, 1, 256, 256) |
| 1 | 1 | Conv2D + ReLU + BatchNorm2D | (7, 7) | (1, 1) | - | (12, 1, 256, 256) |
| 1 | 1 | Conv2D + ReLU + BatchNorm2D | (7, 7) | (1, 1) | - | (24, 1, 256, 256) |
| 2 | 1 | MaxPool2D | (2, 2) | (2, 2) | - | (24, 1, 128, 128) |
| 2 | 2 | Conv2D + ReLU + BatchNorm2D | (7, 7) | (1, 1) | - | (24, 1, 128, 128) |
| $n$ | 1 | MaxPool2D | (2, 2) | (2, 2) | - | $(24, 1, \frac{256}{2^{n-1}}, \frac{256}{2^{n-1}})$ |
| $n$ | 2 | Conv2D + ReLU + BatchNorm2D | (7, 7) | (1, 1) | - | $(24, 1, \frac{256}{2^{n-1}}, \frac{256}{2^{n-1}})$ |
| 8 | 1 | MaxPool2D | (2, 2) | (2, 2) | - | (24, 1, 2, 2) |
| 8 | 2 | Conv2D + ReLU + BatchNorm2D | (7, 7) | (1, 1) | - | (24, 1, 2, 2) |
| 7 | 1 | Upsample2D | - | - | - | (24, 1, 4, 4) |
| 7 | 2 | Conv2D + ReLU + BatchNorm2D | (7, 7) | (1, 1) | - | (24, 1, 4, 4) |
| $n$ | 1 | Upsample2D | - | - | - | $(24, 1, \frac{256}{2^{n-1}}, \frac{256}{2^{n-1}})$ |
| $n$ | 2 | Conv2D + ReLU + BatchNorm2D | (7, 7) | (1, 1) | - | $(24, 1, \frac{256}{2^{n-1}}, \frac{256}{2^{n-1}})$ |
| 1 | 1 | Upsample2D | - | - | - | (24, 1, 256, 256) |
| 1 | 2 | Conv2D + ReLU + BatchNorm2D | (7, 7) | (1, 1) | - | (24, 1, 256, 256) |
| 1 | 1 | Conv2D + ReLU | (1, 1) | (1, 1) | - | (1, 1, 256, 256) |
| 1 | 1 | InputRescaling | - | - | scale: 0.01 | (1, 1, 256, 256) |

planes are chosen randomly at every iteration from the 64 total possible planes, making potentially separate draws of planes for imaging and reconstruction. We show details of all datasets used for training reconstructions in Table 4.

We show the details of our FourierNet2D architecture for training PSFs at the larger field of view in Type A, B, C in Table 12 and our FourierNet3D architecture for training reconstruction networks at the larger field of view in Type A, B, C in Table 13. There are no other networks used for these larger field of view experiments. We also show details for training times for both training PSFs and for training more powerful reconstruction networks in Table 6. All PSFs in these networks were trained for the same number of iterations. However, reconstruction networks for some of these experiments were only trained for as long as necessary to converge (with some exceptions where we attempted longer training to check for performance gains with long training periods). Generally, we observed that performance for such reconstruction networks does not meaningfully change with many more iterations of training[3].

---

[3]Training times are approximate, and actual total time was longer due to checkpointing/snapshotting/validation of data and/or differences in load on the clusters being used.

Table 11: UNet3D detailed architecture (8 GPUs)

| Scale | Repeat | Layer type | Kernel size | Stride | Notes | Shape (C, D, H, W) |
|---|---|---|---|---|---|---|
| 1 | 1 | InputScaling | - | - | scale: 0.01 | (1, 1, 256, 256) |
| 1 | 1 | Conv2D + ReLU + BatchNorm2D | (7, 7) | (1, 1) | - | (30, 1, 256, 256) |
| 1 | 1 | Conv2D + ReLU + BatchNorm2D | (7, 7) | (1, 1) | - | (60, 1, 256, 256) |
| 2 | 1 | MaxPool2D | (2, 2) | (2, 2) | - | (60, 1, 128, 128) |
| 2 | 2 | Conv2D + ReLU + BatchNorm2D | (7, 7) | (1, 1) | - | (60, 1, 128, 128) |
| 3 | 1 | MaxPool2D | (2, 2) | (2, 2) | - | (60, 1, 64, 64) |
| 3 | 2 | Conv2D + ReLU + BatchNorm2D | (7, 7) | (1, 1) | - | (60, 1, 64, 64) |
| 4 | 1 | MaxPool2D | (2, 2) | (2, 2) | - | (60, 1, 32, 32) |
| 4 | 2 | Conv2D + ReLU + BatchNorm2D | (7, 7) | (1, 1) | - | (60, 1, 32, 32) |
| 4 | 1 | Reshape2D3D | - | - | - | (5, 12, 32, 32) |
| 3 | 1 | Upsample2D | - | - | - | (5, 12, 64, 64) |
| 3 | 2 | Conv3D + ReLU + BatchNorm3D | (11, 7, 7) | (1, 1, 1) | - | (5, 12, 64, 64) |
| 2 | 1 | Upsample2D | - | - | - | (5, 12, 128, 128) |
| 2 | 2 | Conv3D + ReLU + BatchNorm3D | (11, 7, 7) | (1, 1, 1) | - | (5, 12, 128, 128) |
| 1 | 1 | Upsample2D | - | - | - | (5, 12, 256, 256) |
| 1 | 2 | Conv3D + ReLU + BatchNorm3D | (11, 7, 7) | (1, 1, 1) | - | (5, 12, 256, 256) |
| 1 | 1 | Conv3D + ReLU | (1, 1, 1) | (1, 1, 1) | - | (1, 12, 256, 256) |
| 1 | 1 | InputRescaling | - | - | scale: 0.01 | (1, 12, 256, 256) |

Table 12: FourierNet2D detailed architecture (1 per plane)

| Layer type | Kernel size | Stride | Notes | Shape (C, D, H, W) |
|---|---|---|---|---|
| InputScaling | - | - | scale: 0.01 | (1, 1, 512, 512) |
| FourierConv2D | (512, 512) | (2, 2) | - | (5, 1, 512, 512) |
| LeakyReLU | - | - | slope: -0.01 | (5, 1, 512, 512) |
| BatchNorm2D | - | - | - | (5, 1, 512, 512) |
| Conv2D | (11, 11) | (1, 1) | - | (1, 1, 512, 512) |
| ReLU | - | - | - | (1, 1, 512, 512) |
| InputRescaling | - | - | scale: 0.01 | (1, 1, 512, 512) |

A.6 DETAILS FOR FOURIERNETS OUTPERFORM STATE-OF-THE-ART FOR RECONSTRUCTING NATURAL IMAGES CAPTURED BY DIFFUSERCAM LENSLESS CAMERA

We performed no augmentations for this set of trainings reconstructing RGB color images of natural scenes from RGB diffused images taken through a DiffuserCam [20]. We modified our FourierNet2D architecture to create the FourierNetRGB architecture and our FourierUNet2D architecture to create

Table 13: FourierNet3D detailed architecture (8 GPUs)

| Layer type | Kernel size | Stride | Notes | Shape (C, D, H, W) |
|---|---|---|---|---|
| InputScaling | - | - | scale: 0.01 | (1, 1, 512, 512) |
| FourierConv2D | (512, 512) | (2, 2) | - | (80, 1, 512, 512) |
| LeakyReLU | - | - | slope: -0.01 | (80, 1, 512, 512) |
| BatchNorm2D | - | - | - | (80, 1, 512, 512) |
| Reshape2D3D | - | - | - | (5, 16, 512, 512) |
| Conv3D | (11, 7, 7) | (1, 1, 1) | - | (5, 16, 512, 512) |
| LeakyReLU | - | - | slope: -0.01 | (5, 16, 512, 512) |
| BatchNorm3D | - | - | - | (5, 16, 512, 512) |
| Conv3D | (11, 7, 7) | (1, 1, 1) | - | (1, 16, 512, 512) |
| ReLU | - | - | - | (1, 16, 512, 512) |
| InputRescaling | - | - | scale: 0.01 | (1, 16, 512, 512) |

Table 14: DLMD experiment training times. Superscripts denote loss function: [1] MSE, [2] MSE+LPIPS.

| Network | Optimizing | # parameters | # train steps | Train step time (s) | Total time (h) |
|---|---|---|---|---|---|
| FourierNetRGB[1] | $\theta$ | $\sim 1.6 \times 10^7$ | $2.2 \times 10^5$ | $\sim 0.43$ | $\sim 26$ |
| FourierNetRGB[2] | $\theta$ | $\sim 1.6 \times 10^7$ | $1.1 \times 10^5$ | $\sim 0.47$ | $\sim 14$ |
| FourierUNetRGB[1] | $\theta$ | $\sim 7.1 \times 10^7$ | $2.5 \times 10^5$ | $\sim 3.3$ | $\sim 229$ |
| Le-ADMM-U[2] [20] | $\theta$ | $\sim 4.0 \times 10^7$ | - | - | - |
| UNet[2] [20] | $\theta$ | $\sim 1.0 \times 10^8$ | - | - | - |

Table 15: FourierNetRGB detailed architecture

| Layer type | Kernel size | Stride | Notes | Shape (N, C, H, W) |
|---|---|---|---|---|
| FourierConv2D | (270, 480) | (2, 2) | - | (4, 3, 270, 480) |
| LeakyReLU | - | - | slope: -0.01 | (4, 20, 270, 480) |
| BatchNorm2D | - | - | - | (4, 20, 270, 480) |
| Conv2D | (11, 11) | (1, 1) | - | (4, 64, 270, 480) |
| BatchNorm2D | - | - | - | (4, 64, 270, 480) |
| LeakyReLU | - | - | slope: -0.01 | (4, 64, 270, 480) |
| Conv2D | (11, 11) | (1, 1) | - | (4, 64, 270, 480) |
| BatchNorm2D | - | - | - | (4, 64, 270, 480) |
| LeakyReLU | - | - | slope: -0.01 | (4, 64, 270, 480) |
| Conv2D | (11, 11) | (1, 1) | - | (4, 3, 270, 480) |
| ReLU | - | - | - | (4, 3, 270, 480) |

the FourierUNetRGB architecture, outlined in Table 15 and Table 16 respectively. Training details are shown in Table 14. Because these reconstructions are of 2D images only and required no microscope simulation, we were able to use a batch size of 4 images per iteration.

Table 16: FourierUNetRGB detailed architecture

| Scale | Repeat | Layer type | Kernel size | Stride | Notes | Shape (N, C, H, W) |
|---|---|---|---|---|---|---|
| 1 | 1 | Multiscale FourierConv2D + ReLU + BatchNorm2D | (270, 480) | (2, 2) | - | (4, 64, 270, 480) |
| 2 | | | (135, 240) | (2, 2) | | (4, 64, 135, 240) |
| 3 | | | (67, 120) | (2, 2) | | (4, 64, 67, 120) |
| 4 | | | (33, 60) | (2, 2) | | (4, 64, 33, 60) |
| 3 | 1 | Upsample2D | - | - | - | (4, 64, 67, 120) |
| 3 | 2 | Conv2D + ReLU + BatchNorm2D | (11, 11) | (1, 1) | - | (4, 64, 67, 120) |
| 2 | 1 | Upsample2D | - | - | - | (4, 64, 135, 240) |
| 2 | 2 | Conv2D + ReLU + BatchNorm2D | (11, 11) | (1, 1) | - | (4, 64, 135, 240) |
| 1 | 1 | Upsample2D | - | - | - | (4, 64, 270, 480) |
| 1 | 2 | Conv2D + ReLU + BatchNorm2D | (11, 11) | (1, 1) | - | (4, 64, 270, 480) |
| 1 | 1 | Conv2D + ReLU | (1, 1) | (1, 1) | - | (4, 3, 270, 480) |

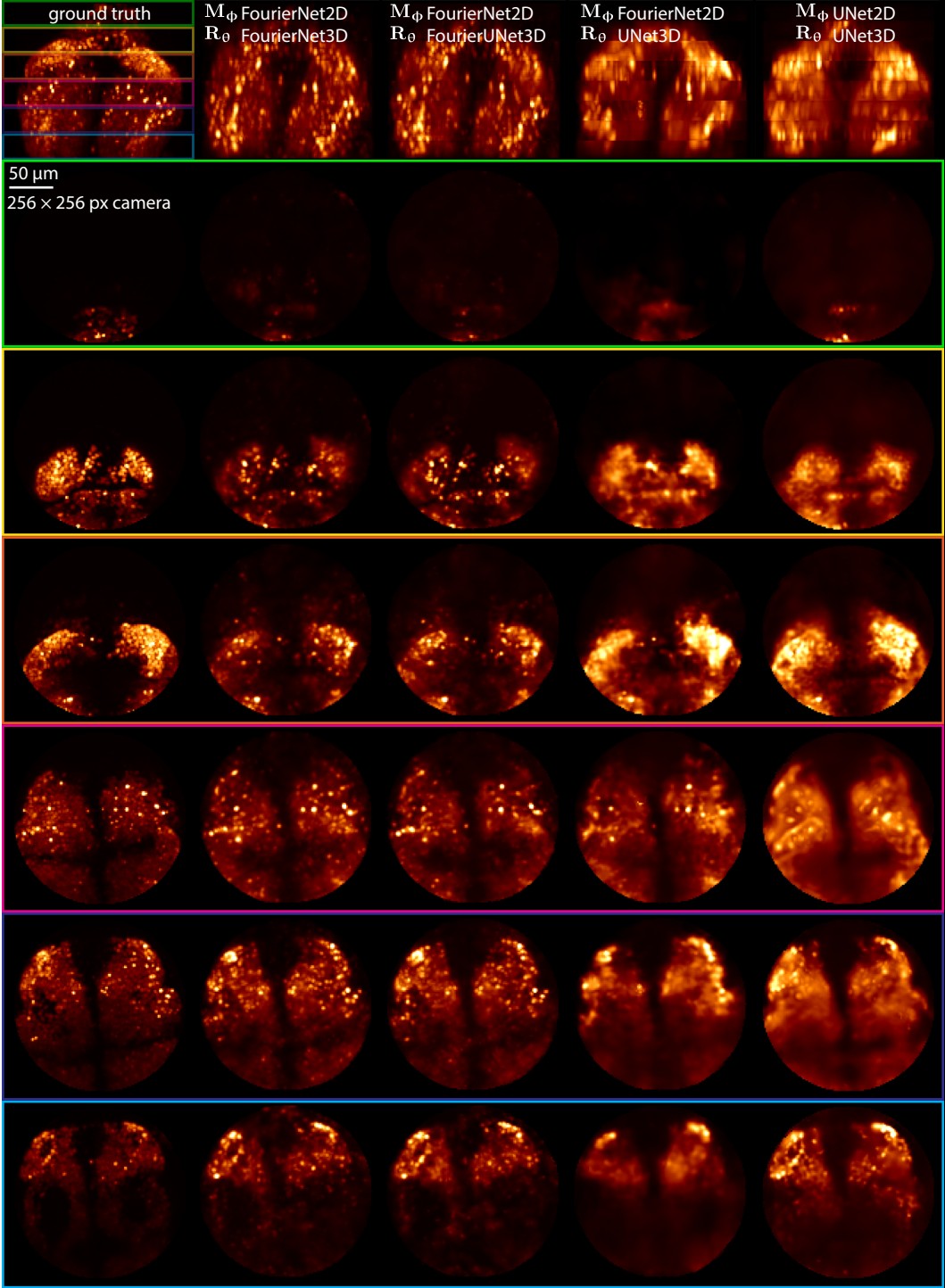

Figure 8: Slab views of a Type D example volume reconstruction, showing our methods (FourierNet/FourierUNet) do the best job of reconstructing throughout the volume. Note that the UNet reconstructions are blurry across all slabs, with few exceptions. Colored boxes show which sample planes a particular slab comes from, corresponding to boxes in xz projection view at top. Annotation $\mathbf{M}_\phi$ shows which network architecture was used for microscope optimization; annotation $\mathbf{R}_\theta$ shows which architecture was used for reconstruction.

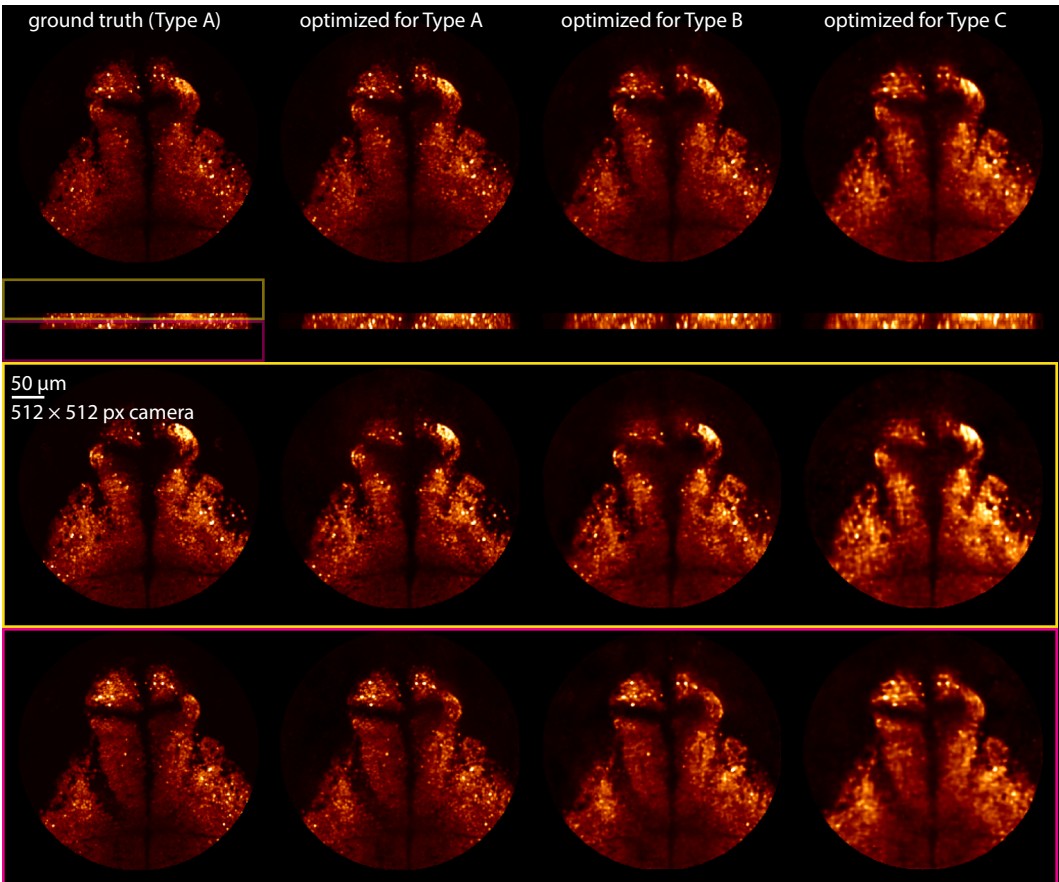

Figure 9: Slab views of an example Type A volume show that the microscope optimized for Type A results in the best reconstructions. Note that the reconstruction with a microscope optimized for Type A is almost identical to the ground truth, while the other microscopes create blurrier reconstructions. Slabs are xy max projections in thinner chunks as opposed to projecting through the entire volume. Colored boxes show which sample planes a particular slab comes from, corresponding to boxes in xz projection view at top.

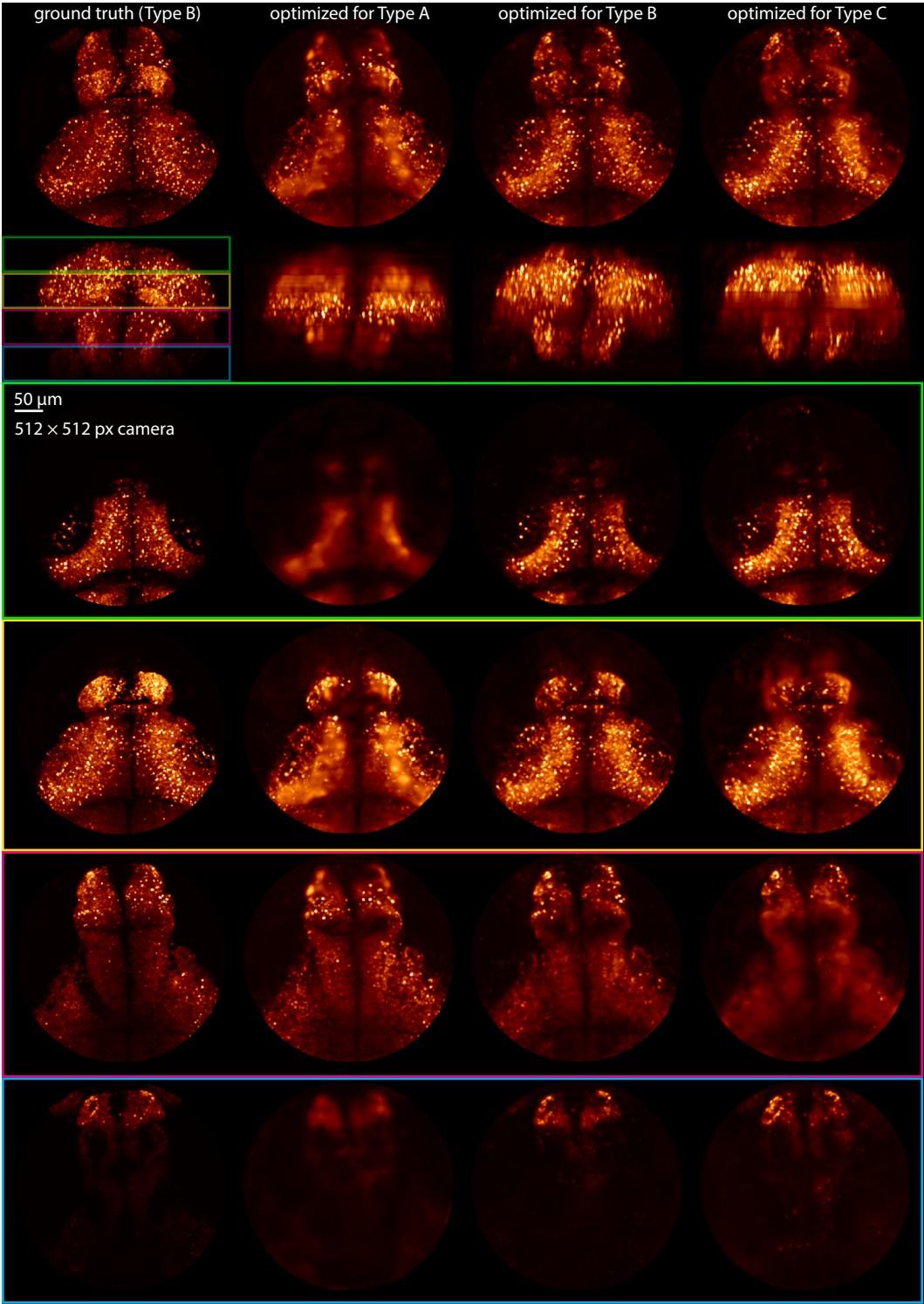

Figure 10: Slab views of an example Type B volume show that the microscope optimized for Type B results in the best reconstructions; other microscopes result in blurrier reconstructions. Colored boxes show which sample planes a particular slab comes from, corresponding to boxes in xz projection view at top.

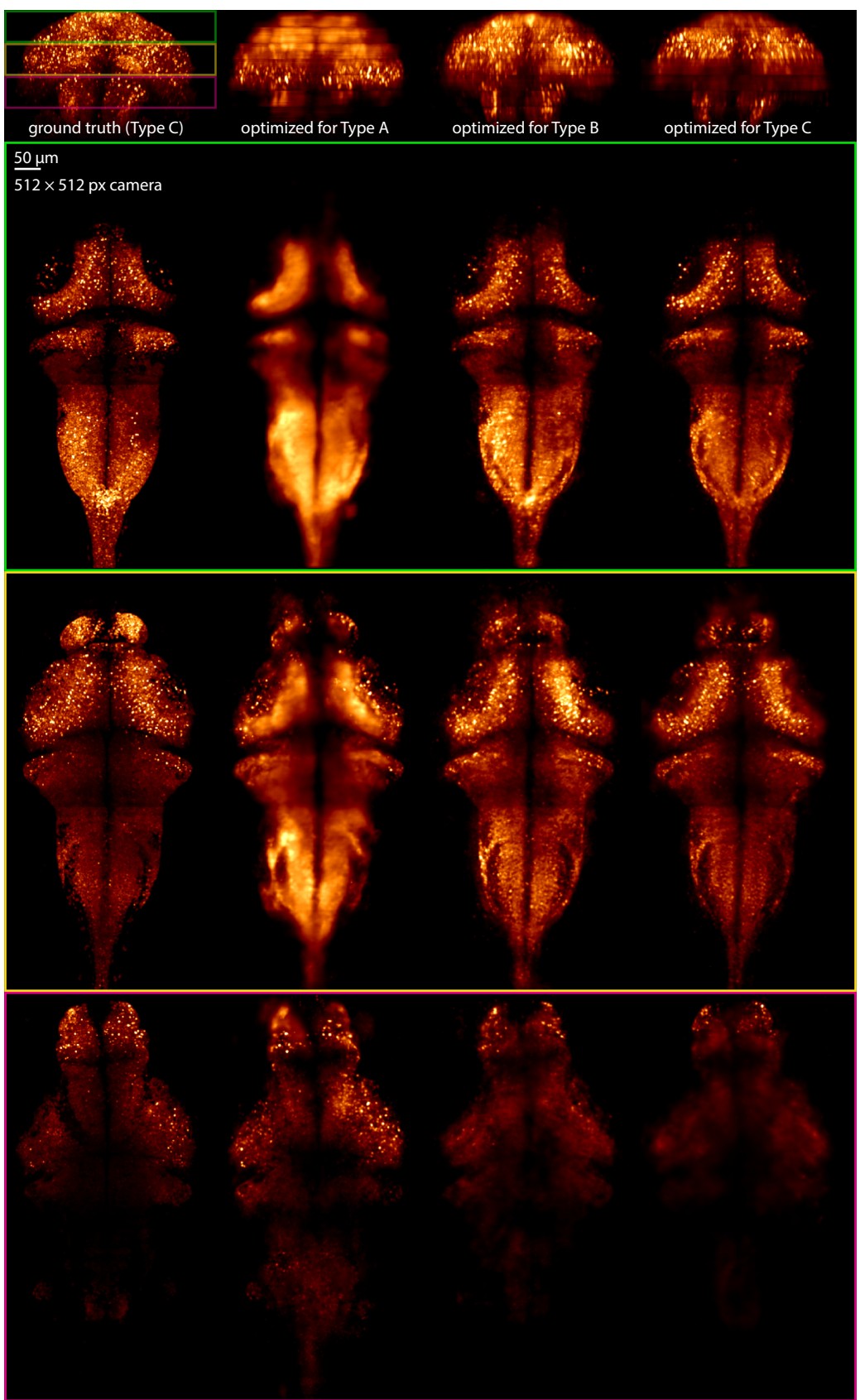

Figure 11: Slab views of an example Type C volume show that microscope optimized for Type C provides most consistent reconstruction. Colored boxes have same meaning as Figures 9, 10.

