# OpenReview forum: "Programmable 3D snapshot microscopy with Fourier convolutional networks"
_ICLR.cc/2022/Conference — ICLR 2022 Submitted_

### Official Review · Reviewer_3cwP · 2021-10-29

**Correctness:** 4
**Technical Novelty And Significance:** 3
**Empirical Novelty And Significance:** 3
**Recommendation:** 6
**Confidence:** 3

**Main Review:**

Positive:
* The paper is nicely written, illustrated. Nice results.
* On the method side, the integration of Fourier-domain layers, while not necessarily 100% novel, seems to do its job quite well and could inspire follow-up work in computational imaging.
* The method is validated against multiple scenarios including an existing database and newly captured data, as well as simpler, more obvious architectures..


Questions:
* I would assume that Fourier transforming the signal is not a new concept in the machine learning community. Among the things that come to mind are papers like Rippel et al., Spectral Representations for Convolutional Neural Networks, NIPS 2015 - but also other interpretations like "positional encoding" (Vasvani et al., Gehring et al., both 2017). Efficient large-kernel convolutions by virtue of the convolution theorem comes up as the main application scenario when doing a web search on "Fourier CNN". I would appreciate if the authors could clarify the relation of this work to those.

Doubt:
* Doing the Fourier convolution in various frequency bands (by cropping) seems somewhat redundant, as the downsampling amounts to cropping in Fourier domain and so one and the same frequency shows up in different layers. Could the desired effect (giving more weight to low frequencies) not be achieved just as well by appropriate frequency-dependent weighting or feature depth?

**Summary Of The Paper:**

This paper introduces a convolutional neural network architecture for designing phase patterns to display on a spatial light modulator in the imaging path of a microscope, as well as decoding the resulting image into a 3D volume. The authors propose to use Fourier features in the decoding stage to enable features that are more strongly nonlocal than in regular (primal-domain) convolution than what they claim would be possible to achieve using "vanilla" UNet/ResNet-like architectures. They demonstrate their approach for snapshot 3D microscopy as well as the reconstruction of light fields from input captured using lensless system (with randomized optical elements).


**Summary Of The Review:**

In my opinion, the foremost contributions of this paper are in the application and the pipeline, and not quite so much in machine learning methodology as the main idea (Fourier convolution) has been around for quite some time. The proposed pipeline is thoroughly motivated, described and evaluated. To me, this looks like an acceptable paper.

That said, I'm not an expert on the application (though it does feel like a bit of a niche). My relation to machine learning is that of a casual user rather than a contributor of hardcore methodology. So I'm looking forward to reading the other opinions on the work.

---

> ### Author Response · Authors · 2021-11-21
> **Response to Reviewer 3cwP**
>
> Thank you for your attentive and favorable review.
>
> We would like to clarify that our main contribution is demonstrating that FourierNets enable end-to-end optimization of highly non-local optical encoders as well as high quality reconstructions for computational imaging applications. We apologize for conveying the impression that we are the first to describe Fourier domain convolution layers, and we have included a citation to Rippel 2015. Further, the idea to use large kernels (without Fourier convolution) is also not a new idea in computer vision (CVPR Sun 2017, Zhang 2019) and we have included these citations as well.
>
> We agree that simply weighting different frequency bands could achieve the same effect as cropping. Our original FourierNet does indeed see all frequencies, and our cropping of Fourier features in FourierUNet is essentially forcing different network paths to ignore certain frequencies. What we wanted to show with FourierUNet is the effect of removing the UNet-style encoder while keeping the UNet-style decoder. This allowed us to investigate whether the UNet-style multiscale decoder is useful for the optical problems we are examining or whether simply having a global convolution is the more important factor.
>
> Finally, we want to mention that newly available programmable optical elements combined with modern machine learning techniques have the potential to revolutionize computational imaging. Understanding how to optimize these programmable elements with over 10^6 parameters effectively will become increasingly important. We think this new research area of programmable computational microscopy is poised for major growth.

---

> > ### Comment · Reviewer_3cwP · 2021-11-29
> > **Thank you**
> >
> > Dear Authors,
> >
> > Thank you for addressing my concerns. Your rebuttal confirms me in my assessment that this is acceptable work.
> >
> > Kind regards

---

### Official Review · Reviewer_jREz · 2021-10-31

**Correctness:** 3
**Technical Novelty And Significance:** 2
**Empirical Novelty And Significance:** 3
**Recommendation:** 6
**Confidence:** 4

**Details Of Ethics Concerns:**

I believe that there is no problem with ethics.

**Main Review:**

This paper is well written with a clear description. I like the idea of incorporating Fourier convolutional neural networks to more efficiently propagate gradients to the global structures, which allow for handling large PSFs often desired in high-dimensional imaging. Although the proposed architectures are simple modifications to the existing CNNs, their motivation is clear and demonstrates the advantages in synthetic experiments. In addition, all the evaluations are pretty well designed and executed.

Nevertheless, I have some questions and concerns about the claimed contributions shown in Section 1.2.

First, the authors highlight that the parallel differentiable wave optics simulation achieved the optimization of 10^6 phase-modulating pixels. It is not clear whether achieving 10^6 pixels as optimization variables can be only achieved using the parallel simulator. Also, having 10^6 pixels is necessary to achieve the presented performance? In other words, I wonder how the performance of the proposed method would vary as the pixel count changes. Even though this is often regarded as a training detail in most end-to-end optimization papers, validating this would make the main contribution of this paper clear to readers because it could explain the motivation for the main contribution.

The second concern is, probably as expected by the authors, is the missing real-world experiments. The domain gap between the synthetic and real experiments is generally large for the end-to-end optimization of optics and reconstruction, making it challenging to incorporate many optimization variables for the optics encoder. This means that directly applying the proposed high pixel-count optics optimization to real-world experiments may impose challenges partially conflicted with the core idea of increasing optimization parameters in this work. I do not believe that this question comes as a surprise to the authors, but a more detailed discussion on this potential problem is worth to be given in the paper.

**Summary Of The Paper:**

This paper proposes a neural network architecture that applies a series of convolution kernels on the Fourier domain to focus on global information of the captured images. The authors make use of the Fourier-plane phase modulation and apply this idea to single-shot 3D microscopic imaging using the Fourier-plane phase modulation for 3D microscope and lensless imaging. Also, this is further extended to the end-to-end optimization of the phase modulation and the reconstruction network. An extensive synthetic evaluation is conducted to validate the effectiveness of the proposed method over existing non-Fourier neural networks in the context of computational imaging.

**Summary Of The Review:**

In summary, there are some major issues better be addressed in the paper. However, I don't see them as a critical barrier for blocking this paper from being accepted, given its high-quality execution and potential impact on computational imaging research.

---

> ### Author Response · Authors · 2021-11-21
> **Response to Reviewer jREz**
>
> Thank you for your insightful and favorable review. We hope to clarify some of your concerns.
>
> 1a. The need for a parallel simulator
>
> Memory usage in our simulation scales strongly with the field of view (FOV) and number of planes being imaged, and only weakly with the number of pixels in the phase mask. For smaller FOVs, the optimization can indeed be performed on a single GPU. For our large FOV experiments (Type A, B, C), each plane requires approximately 1.3 GB of additional memory when optimizing both reconstruction networks and phase masks. In order to optimize a reasonable number of planes (e.g. 64 for these experiments), multiple GPUs will be required.
>
> 1b. The need for 10^6 pixels in the phase mask
>
> The field of view that we plan to image determines the number of pixels required in the phase mask. This is a result determined by the physics of the microscope. For a microscope with a 0.8 NA objective, and field of view of 832 microns, we need 2560 x 2560 pixels in the phase mask (spatial light modulator).
>
> 2. Regarding the gap between simulation and reality
>
> The main contribution of our paper is describing the solutions to two key bottlenecks to the simulation-based optimization of the parameters of optical systems with highly non-local PSFs: efficient and effective reconstruction neural networks, and large-scale multi-GPU distributed differentiable simulations. These technical contributions do not require any experimental validation.
>
> However, the engineered phase masks resulting from our optimization have not yet been experimentally validated. Note that these phase masks should not be considered the main contributions of our paper. Even so, we address a few potential concerns regarding the implementation of a 3D snapshot microscope as simulated in our paper. First, we should point out that spatial light modulators with greater pixel counts than 2560x2560 are now readily available (e.g. Thorlabs EXULUS-4K1 or Holoeye GAEA-2). Second, the SLM can be used to simultaneously correct for system aberrations and implementing our target point spread function (optical encoding). Small imperfections in the implementation can still be measured perfectly and used to train reconstruction networks to near optimal performance.

---

### Official Review · Reviewer_nest · 2021-11-02

**Correctness:** 3
**Technical Novelty And Significance:** 3
**Empirical Novelty And Significance:** 2
**Recommendation:** 6
**Confidence:** 3

**Main Review:**

#### Strengths
* The idea of using Fourier transform to achieve large-size PSF is interesting.
* The design of the Fourier network is well explained: There are some features to reduce the computation in the Fourier network (e.g., save the parameters in Fourier space to save a padding operation) and they are explained in detail. The explanation of the multi-scale features in the Fourier domain is also helpful to understand the effectiveness of the proposed architecture.
* The paper has a nice section on limitations.

#### Weaknesses
* Lack of evaluation: While this paper introduces an interesting architecture, the results are not sufficient to demonstrate the effectiveness of the architecture. First, the 3D snapshot microscope is evaluated only on simulated data. As the author mentioned in the limitations paragraph, the real optical encoder will have some artifacts because of aberrations and calibration error, and these effects need to be evaluated. Second, the evaluation is only on one type of volume (i.e., Larval Zebrafish). While the paper shows additional results on lensless imaging, it cannot support 3D snapshot microscopy, which is the title of this paper.
* The result from UNet is unclear to me. Although it is computationally inefficient, UNet should be able to achieve global context if it has enough layers and thereby produce similar quality of results. However, in Figure 2, UNet provides much worse results compared to FourierNet. Why does UNet provide such bad results? Is it because the parameters of UNet are not sufficient to handle the global context?
* Table 3: Why is UNet much faster compared to the FourierNet results? This is the opposite of what is claimed in the paper. Also, why is the FourierUNet result missing in this experiment?
* I understand Lensless imaging results are included to demonstrate the effectiveness of Fourier network, but it is somewhat unnatural to have this new application abruptly considering the title of the paper starts with “3D snapshot microscopy”

#### Minor comments
* Figure 2: What is the 3.00s in the ground truth of the second row?

##### Typos
* “field field” in the 3rd paragraph of A.1
* Table 3: unnecessary line break in the caption
* Table 4: unnecessary line break in the second row.


**Summary Of The Paper:**

This paper introduces an end-to-end optical encoder and deep learning decoder optimization for 3D snapshot microscopy. The main challenge is that the decoder needs to be able to handle the global PSFs used in the optical encoder in 3D snapshot microscope. This is difficult to be achieved in conventional UNet architectures as the global context requires many layers that is computationally expensive. The authors use a Fourier convolutional network that achieves the global context in a single layer with a much lower computational cost. The method is demonstrated on simulated data of one volume with four different fields of view. The authors also applied the technique to lensless imaging.


**Summary Of The Review:**

This paper proposes an interesting architecture for the end-to-end optimization of the optical encoder and deep learning decoder for 3D snapshot microscopy. In particular, the authors introduce the Fourier network to handle the global PSF in the encoder with a lower computational cost compared to conventional UNet architecture. However, the method is evaluated only on simulated data, with a single type of object, and the explanation of conventional UNet result is insufficient to demonstrate the effectiveness of the proposed architecture. So the decision is weak reject.

---

> ### Author Response · Authors · 2021-11-21
> **Response to Reviewer nest**
>
> We appreciate your thoughtful review and hope to clarify some of your concerns.
>
> Our main contribution is a framework for simulation-based optimization of highly non-local PSFs for 3D snapshot microscopy. We solved two challenges limiting existing PSF optimization approaches, by (1) developing a distributed and differentiable multi-GPU wave-optics simulation of the microscope, and (2) developing a FourierNet based architecture for efficiently and accurately reconstructing 3D volumes from the snapshot images.
>
> Previous approaches only optimized small local PSFs for computational reasons and, as we show, due to implicit biases of the UNet architecture. Even though the UNet has global context, our results show it is still biased towards using local information for its predictions. This bias leads to learning local optical encodings/PSFs when performing end-to-end learning with UNet decoders. PSF optimization is a highly non-convex problem, with many local minima. UNet decoders appear to find rather bad local minima. Our contribution is showing that FourierNets achieve far superior local minima due to the lack of a locality bias.
>
> We also want to demonstrate that FourierNets are not just good for end-to-end learning, but are effective for decoding without training the optical encoder. For this reason, we also showed that FourierNets are superior to UNets not only for our case of 3D snapshot microscopy, but also for lensless photography. The lensless photography reconstruction problem has been well studied, with a variety of reconstruction algorithms. Thus, it poses a challenging application, and the fact that our FourierNets outperform all previous methods demonstrates their general utility.
>
> 1. Lack of evaluation, only demonstrated on 1 volume
>
> We demonstrated the simulation-based optimization of nonlocal PSFs for the problem of 3D snapshot microscopy. This is a challenging application requiring significant effort to simulate. We collected a large database of real zebrafish images exclusively for our paper, requiring several person-months. This unique, large dataset of high-resolution confocal ground truth images of the entire zebrafish nervous system (58 training fish, 10 test fish) is challenging to replicate and will be released on acceptance. While our figures show a single volume for convenient comparison, we always evaluate on the entire test set.
>
> Using our ground truth zebrafish volumes, we performed simulation-based optimization of microscope parameters for several different experimentally relevant settings. Beyond demonstrating the effectiveness of our general end-to-end learning framework, we explored the potential of programmable microscopy to optimally tailor optical encodings to different experimental settings: (Type A) imaging a thin section of the 3D volume at high resolution and speed, (Type B) imaging just the forebrain of the zebrafish, and (Type C) imaging the entire brain of the zebrafish. We hypothesized that there would be trade-offs in these three situations: the best setting for imaging Type A samples would not be the same as the best setting for Type C samples. Through our simulation-based optimization, we demonstrated that this is indeed the case.
>
> We believe our experiments have effectively demonstrated our main claims: that FourierNets are superior to UNets for both end-to-end PSF optimization and for image reconstruction, and that programmable microscopy can enable optimal imaging for different imaging conditions.
>
> Finally, we note that PSF optimization can only be performed in simulation. Since our focus is developing a new simulation-based PSF optimization framework, we believe we can demonstrate our claims purely in simulation.
>
> 2. UNet results
>
> We made sure that the UNet receptive field was global for 256x256 pixels input size. Despite this, reconstruction results with a UNet-optimized PSF are significantly worse than with a FourierNet-optimized PSF, shown in Table 1 and Figure 2, due to the locality bias discussed previously. Table 5 in the supplement shows parameter counts for all our snapshot microscopy networks, and we can see that for reconstruction, UNets actually have more parameters than FourierNets; for PSF optimization, UNets and FourierNets have comparable parameter counts.
>
> 3. Table 3
>
> UNet is faster because the reconstruction is for a single RGB image, so the small UNet with 3x3 convolutions is fastest. We have updated the language so we are not claiming FourierNets are always faster, because this depends on network size. FourierUNet achieves an MSE of 0.43 (x10^-2), an LPIPS of 0.22, an SSIM of 0.875, and a PSNR of 24.5. We have included these results in Table 3.
>
> 4. Figure 2
>
> The 3.00s in Figure 2 shows the estimated acquisition time, which is much higher for confocal ground truth than for the reconstructions (scanning multiple planes for confocal versus a single image for snapshot). We have further clarified this in the caption for Figure 2.

---

> > ### Comment · Reviewer_nest · 2021-11-29
> > **Response to Authors**
> >
> > Dear Authors,
> >
> > Thank you for the response. It clarified the detail of the simulated dataset and addressed the concern on the evaluation. The update of Table 3 also addressed some of my concerns though the speed issue should be explained more carefully as it depends on network size as you mentioned. I still think the evaluation can be improved (e.g., lensless imaging experiment does not fully demonstrate the FourierNet in 3D snapshot microscopy because of different implementation details as in speed comparison) but it appears to be reasonable considering the difficulty of obtaining real data. I improve my rating to 6.

---

### Official Review · Reviewer_cUSr · 2021-11-03

**Correctness:** 4
**Technical Novelty And Significance:** 2
**Empirical Novelty And Significance:** 3
**Recommendation:** 6
**Confidence:** 3

**Main Review:**

Strengths
-------------
- The paper is well written and easy to understand.
- The experimental results are promising.
- I really appreciate the thorough limitations section.


Weaknesses
-----------------
- The paper has limited technical novelty when compared to a previous paper which implemented Fourier-domain networks, and does not currently contain experiments showing that the differences in approaches provide a meaningful improvement (see Detailed Comments).
- The paper is missing some implementation details which affect the reader’s understanding of the Fourier structure of the neural network (see Detailed Comments).

Detailed Comments
--------------------------
- The following paper contains very closely related work and should be cited and discussed (I am not affiliated with this paper):

    Rippel, Oren, Jasper Snoek, and Ryan P. Adams. "Spectral representations for convolutional neural networks." Proceedings of the 28th International Conference on Neural Information Processing Systems-Volume 2. 2015.

    In particular, that paper proposes to learn multiplicative weights in Fourier space instead of learning image space convolutional kernels. The learned weights are parameterized to correspond to convolutional kernels with limited extent in the image space, which outperform image-space convolutions on a CIFAR classification task (Fig. 5a). This paper does differ from Rippel et al. in that the the learned convolutions have a kernel the size of the full image. It thus seems necessary to show that it is the global extent of the filters in this paper, and not the fact that the optimization is being performed in the space of the multiplicative Fourier weights, is driving the performance improvements. Further, that paper previously introduced the spectral cropping approach described in this paper for doing multi-scale learning in the Fourier space (although this is used for a classification task and not in the context of a U-net architecture).

- I believe that the input images and output images of the networks are real-valued, and thus have a complex-valued Fourier transform with conjugate symmetry. (I do not have experience with this type of imaging and am going off Eq. 12-17). I see that the learned weights are complex valued.  How is the real valued-ness of the output of each layer maintained? Are the multiplicative weights chosen such that the output in Fourier space has conjugate symmetry? Since the core contribution of this paper is the Fourier network, these details would be useful to include in the main text, along with the fact that the multiplicative weights are complex-valued -- I think this is currently only mentioned in the Appendix.

- Could you provide a more concrete sense of the memory requirements of the FourierNetRGB Architecture as compared to those of Le-ADMM-U and/or a comparison of the number of parameters? I see these for the other Fourier architectures but not FourierNetRGB in the Appendix. If the memory requirements/number of parameters is significantly smaller for Le-ADMM-U, a comparison may be needed with a version of that architecture scaled to match for fair comparison. I understand for the microscopy experiments that the imaging simulation requires significant memory, which is what motivates the multi-GPU implementation. However, it’s hard to know concretely what that means for the lensless imaging comparison, since only reconstruction is being performed in this case, and it would be good to know that the comparison in Table 3 is fair.

    Relatedly, could you provide the total training times for the architectures in the lensless imaging example? I assume the times in Tables 2 and 3 are the total inference time for reconstructing a single image; it is good to see that FourierNet is faster than Le-ADMM-U^2 there, but I don’t have a sense of whether the 20ms difference is relevant for a practical application. It would be great for readers to know whether there are significant differences in training time when choosing between these methods.

**Summary Of The Paper:**

This paper introduces the use of Fourier convolutional neural networks to model both image encoding and reconstruction processes. These neural networks learn multiplicative weights in the Fourier domain, enabling efficient global convolutions (i.e., convolutions with kernels the size of the entire image), which are particularly important for imaging applications with a wide PSF. These networks compare favorably to UNets for the task of joint 3D snapshot microscope parameter optimization and image reconstruction. These networks also outperform the state-of-the-art architectures for a lensless imaging reconstruction task.

**Summary Of The Review:**

This is a well-written paper with good reconstruction results. In its current form, the paper lacks important comparisons/contextualization with a very closely related work (Rippel et al.), which previously introduced many of the main technical ideas in this in this work. Also, for the lensless imaging task in this paper, more details are needed to understand whether the resources required for this method and the state-of-the-art are comparable.

Overall, I would be inclined to improve my rating if the paper is extended with (1) a more careful comparison with Rippel et al., demonstrating which specific aspects of this work provide innovation and improvement over previous approaches and (2) more details about the resources required for the lensless imaging experiments, to better understand whether the comparison provided is fair.

---

> ### Author Response · Authors · 2021-11-21
> **Response to Reviewer cUSr**
>
> Thank you for the careful review. We would like to address the main points that you brought up.
>
> Our main contribution is demonstrating that FourierNets enable the end-to-end optimization of highly non-local optical encoders, and for high quality reconstructions for two computational imaging applications. As you point out, we are not the first to describe convolutions filters parameterized in the Fourier domain, and we apologize for conveying this impression. We have now included citations to Rippel 2015, and also other previous works using non-Fourier domain large filter convolutions for other computer vision applications such as semantic segmentation and salient object detection (Sun 2017, Zhang 2019). Further, we agree with you that the findings in Rippel 2015 of the optimization benefits of parametrizing the filters in the Fourier domain also apply to our work.
>
> Finally, we thank you for some of the more technical points. The input and output of our Fourier convolution layers are indeed real valued. Since imposing conjugate symmetry in the weights comes at a computational expense and no change in expressivity, we chose to use the overparametrized form. To obtain a real value from the Fourier convolution, we simply keep only the real part of the result after the inverse Fourier transform. Regarding number of parameters for the DiffuserCam experiments, our FourierNetRGB model in that setting had 1.6x10^7 parameters, as compared to 1.3x10^7 parameters for Le-ADMM-U and 1.2x10^7 parameters for their UNet (we consider these to be comparable numbers of parameters that would not explain the large differences in quality across the three methods). In this setting, there is no simulation required because the images have been captured on a real lensless camera prototype; the memory usage is solely due to the reconstruction/optimization of the reconstruction network. Because these are results from Monakhova 2019, we do not know how long they trained for. However, we can say that we trained for 2.2x10^5 iterations (with a batch size of 4 images per training iteration) and this took approximately 26 hours of training. We have included this information in a supplementary table (Table 14).

---

> > ### Comment · Reviewer_cUSr · 2021-11-21
> > **Response to Response to Reviewer cUSr**
> >
> > Thanks to the authors for their response to my review -- I appreciate their incorporating the reference to Rippel (and the other works) into the paper. And I appreciate the additional implementation details; this improves my confidence that the provided comparisons are fair.
> >
> > I am improving my rating to 6: marginally above the acceptance threshold. This is a good paper that contains thorough, high-quality application-specific work. However, in its present form, I am not sure that ICLR is the right venue for it; all of the main machine learning ideas in this paper have already been deeply explored in the Rippel paper. I think this paper would shine in an application-specific conference or journal, where it would be seen by more people who can build off and improve on the ideas here.
> >
> > I am still open to changing this review, especially if I have missed a reason that there are new ML ideas in this paper, if the AC/other reviewers feel that this particular application is of specific interest to ICLR, or if one of the other reviewers has outstanding concerns with the experiments (although I have read the rebuttals and am currently satisfied with the experiments).

---

> > > ### Author Response · Authors · 2021-11-23
> > > **Discussion with Reviewer cUSr**
> > >
> > > Thank you for your consideration and for revising your score. We would like to address your remaining concern.
> > >
> > > Our paper is, as you note, an application-focused paper. The ICLR call for papers solicits research with “applications in audio, speech, robotics, **neuroscience**, computational biology, or **any other field**.” The call for papers also states that ICLR considers “a broad range of subject areas including … applications in **vision** … and **others**.” Our paper describes an exciting project at the intersection of (deep) (model-based) machine learning, differentiable physics simulation, computational optics, and neuroscience. We believe that ML-powered imaging is an important emerging research application in the ML community, and believe it will have a broad and diverse readership.

---

### Decision · Program_Chairs · 2022-01-20

**Decision:**

Reject

**Comment:**

The paper proposes an application of the CNN to the microscopy problem of constructing 3D volumes from 2D captured images. The four reviewers thought the paper was a straightforward application of existing techniques to a new problem, while they were borderline towards accept the overall sentiment was that the technical novelty was very low from an ML perspective, and the ICLR community would only find the application potentially of interest. (Two reviewers changed from borderline reject, while the other two chose not to change their scores following the authors’ request.)